# Piezo2 regulates colonic mechanical sensitivity in a sex specific manner in mice

Jonathan Madar[1], Namrata Tiwari[1,2], Cristina Smith[1,2], Divya Sharma[1], Shanwei Shen[1], Alsiddig Elmahdi [1] & Liya Y. Qiao [1] ✉

The mechanosensitive ion channel Piezo2 in mucosa and primary afferents transduces colonic mechanical sensation. Here we show that chemogenetic activation or nociceptor-targeted deletion of Piezo2 is sufficient to regulate colonic mechanical sensitivity in a sex dependent manner. Clozapine N-oxide-induced activation of Piezo2;hM3Dq-expressing sensory neurons evokes colonic hypersensitivity in male mice, and causes dyspnea in female mice likely due to effects on lung sensory neurons. Activation of Piezo2-expressing colonic afferent neurons also induces colonic hypersensitivity in male but not female mice. Piezo2 levels in nociceptive neurons are higher in female than in male mice. We also show that Piezo2 conditional deletion from nociceptive neurons increases body weight growth, slows colonic transits, and reduces colonic mechanosensing in female but not male mice. Piezo2 deletion blocks colonic hypersensitivity in male but not female mice. These results suggest that Piezo2 in nociceptive neurons mediates innocuous colonic mechanosensing in female mice and painful sensation in male mice, suggesting a sexual dimorphism of Piezo2 function in the colonic sensory system.

A myriad of physical forces including osmoregulation, shear, pressure, and repetitive deformation occur during colonic fluid balance, motility and peristalsis. Physical forces against the gastrointestinal (GI) wall can be transduced by mechanosensory afferents of dorsal root ganglia (DRG) neurons which convey sensory information to the central nervous system[1,2]. The elevated activity of mechanosensory afferents enhances mechanosensation of the colon, which underlies a number of GI disorders, such as colonic pain hypersensitivity in irritable bowel syndrome (IBS) and inflammatory bowel diseases (IBD)[3–6]. The Piezo channels, Piezo1 and Piezo2 in most vertebrates, that are discovered by Dr. Ardem Patapoutian for winning the Nobel Prize in Physiology or Medicine 2021, are long-sought-after mechanotransducers that convert internal and external physical forces into chemical or electrical signals to further trigger neuronal action potentials in touch and pain sensation[7–9].

Piezo2 (encoded by *Fam38b*) in DRG is expressed by low-threshold mechanoreceptors (LTMRs) essential for touch sensation, by parvalbumin+ neurons engaging in proprioception, and by nociceptive neurons[10–18] marked by calcitonin gene-related peptide (CGRP)[15] and/or sodium channel Nav1.8[16,19]. Patients with inherited Piezo2 loss-of-function have general losses in vibration detection, touch discrimination, joint proprioception, and deficient bladder-filling sensation[20–22]. Conditional deletion of Piezo2 from sensory neurons in mice leads to decreased neuronal responses to lung inflation[23], reduces somatic sensitivity to light-touch stimuli of the paws and cornea[24], and impairs bladder voiding function[22]. When Piezo2-expressing sensory afferents are optogenetically activated, it is sufficient to initiate baroreflex[25] and induce somatic nociception[26]. Moreover, fly Dm*piezo* knockout from sensory neurons severely reduces behavioral responses to noxious mechanical stimuli[27]. Under pathophysiological states such as inflammation or nerve injury, Piezo2 genetic knockout or antisense RNA knockdown from sensory neurons impairs nocifensive responses to somatic and visceral mechanical stimulation, suggesting the role of Piezo2 in mechanical allodynia[26,28,29].

The Piezo channels in the bowel are identified in mechanosensitive enteric neurons (MEN)[30–32] and epithelial enterochromaffin

[1]Department of Physiology and Biophysics, Virginia Commonwealth University, Richmond, VA, USA. [2]These authors contributed equally: Namrata Tiwari, Cristina Smith. ✉e-mail: liya.qiao@vcuhealth.org

cells[8,33–35] that are suggested to have roles in colonic mechanosensing. Whilst the role of Piezo2 in primary afferent neurons in regulation of colonic mechanosensing and pain hypersensitivity has not been examined. Here, we utilized a variety of transgenic mouse lines and applied a colonometry technique that was developed in our lab to objectively measure colonic mechanical sensitivity in free-moving healthy and diseased mice[36]. We observed a previously uncharacterized phenomenon that Piezo2 in sensory neurons of female mice was more profound to maintain the homeostasis of colonic mechanosensing than in male mice. While in colonic hypersensitivity, Piezo2 had a more prominent role in male mice than in female mice. These findings suggest a sexual dimorphic role of Piezo2 in colonic mechanosensing and pain hypersensitivity.

## Results

### Sexual biases of chemogenetic activation of Piezo2-expressing cells in facilitation of colonic mechanical hypersensitivity

Piezo2 was identified in the enterochromaffin (EC) cells to transduce mechanical energy into chemical signals[8,33,34]. We found that Piezo2 was also expressed in colonic afferent neurons, which was confirmed by utilizing Piezo2;GFP mice (Fig. 1a, Fig. S1a) and determining Piezo2 immunoreactivity (IR) (Figs. S1a, 1b) in combination with conventional neuronal tracing dye Fast Blue (FB) retrograde labeling of colonic primary afferents in DRG (Figs. 1a, b: blue cells). Piezo2;GFP was expressed in peptidergic DRG neurons identified by CGRP-IR (Fig. S1b: indicated by white arrows), primary afferent nerve fibers in the distal colon marked by CGRP (Fig. S1c: indicated by white arrows) or substance P (Sub P) (Fig. S1d: indicated by white arrows). Piezo2;GFP in the distal colon was present lightly in the serous membrane (Fig. 1c: indicated by yellow arrows), strongly in epical mucosal cells (Fig. 1c: indicated by white arrows), and few in enteric nervous system (Fig. S1e: co-localization with protein gene product (PGP)9.5 indicated by red arrows), which were consistent to previous findings showing Piezo2 rarely in enteric neuronal soma but in few neurites[30]. Therefore, Piezo2 was expressed in a variety of regions in the GI reflex pathway, including specifically labeled colonic afferent neurons and sensory fibers.

To examine the role of Piezo2 in regulation of colonic mechanosensing, we implemented Designer Receptors Exclusively Activated by Designer Drugs (DREADD)−based Gq-mediated activation of Piezo2-expressing cells in Piezo2;hM3Dq mice that we generated. We performed colonometry to objectively and quantitatively characterize colonic mechanical sensitivity[36]. We validated hM3Dq (mCherry) expression to be in DRG neurons (Fig. 1d, Fig. S2a: red cells), multiple regions in the distal colon (Fig. 1e, Fig. S2b: red cells) with clear visibility in mucosal cells morphologically resembling the EC cells (Fig. 1e, S2c: indicated by white arrows), however, very scarce in the spinal cord showing no clear red (mCherry) cells in the laminae I-II where sensory nerve central terminals innervated (Fig. 1f, Fig. S2d), which were consistent to previous findings[23]. We injected clozapine N-oxide (CNO) intraperitoneally (i.p.) at 3 mg/kg body weight (this dose was customized through our previous studies[36]) to Piezo2;hM3Dq male mice to activate Piezo2-expressing cells, which increased the amplitude of intracolonic pressures ($A_{ICP}$) within 1-2 h post CNO injection, suggesting an enhanced colonic mechanical sensitivity[36] when compared to the same animal prior to CNO injection (Fig. 1g). Wildtype (wt);hM3Dq male mice that received the same dose of CNO treatment did not demonstrate colonic mechanical hypersensitivity[36], suggesting that CNO-induced colonic hypersensitivity was a result of hM3Dq-initiated activation of Piezo2-expressing cells, excluding the off-targeted responses.

We performed parallel studies in female Piezo2;hM3Dq mice. To our surprise, the CNO dose that was safe for male Piezo2;hM3Dq mice (1-3 mg/kg body weight, i.p.) was lethal for female Piezo2;hM3Dq mice. We lowered the dose of CNO injection and found that a dose of CNO higher than 10 μg/kg body weight (i.p.) also induced dyspnea in female

Piezo2;hM3Dq mice, regardless of their estrus stages. The mice did not recover within 2-4 h of active observations (6 mice were tested for a variety of doses). Interestingly, female wt;hM3Dq mice that did not have Cre-based expression of hM3Dq did not demonstrate the characteristics as female Piezo2;hM3Dq mice to be vulnerable to CNO treatment (more than 3 mice were tested). We also tested the effects of CNO (1-3 mg/kg body weight i.p.) on other female mice that carried hM3Dq expression under other promoter-regulated Cre-driven recombination, and they all survived well after CNO treatment (more than 3 mice tested). These findings suggest that Gq-mediated activation of Piezo2-expressing cells disturbed homeostasis of female mice and threatened their survival.

We next injected CNO intrathecally (i.t.) to Piezo2;hM3Dq mice to examine whether Piezo2 cell-mediated colonic mechanical hypersensitivity involved Piezo2 expression in DRG (Fig. 1h). This approach of localized application of CNO was successfully used to target brain cells specifically and we adapted the suggested concentration of CNO[37] at a dose of 3 μL of 26 μM CNO solution. In response to CNO i.t. injection, Piezo2;hM3Dq male mice demonstrated colonic mechanical hypersensitivity (Fig. 1h) when compared to colonic sensitivity in saline i.t. controls. In a separate group of Piezo2;hM3Dq male mice, we recorded the activity of the urinary bladder by cystometry prior to (Fig. S3a) and post CNO injection (i.t.) (Fig. S3b) in the same animal. CNO injection greatly reduced the inter-micturition intervals (Fig. S3c), suggesting that activation of Piezo2-expressing DRG neurons also led to bladder hyperactivity. Moreover, Piezo2;hM3Dq male mice demonstrated sensory hypersensitivity tested by gait assay (Fig. S3d) which showed decreases in stride length following CNO injection (i.t.) when compared to saline i.t. injection (Fig. S3e). When we applied CNO i.t. injection to female Piezo2;hM3Dq mice, we noticed again that the dose of CNO applied to male Piezo2;hM3Dq mice was toxic to female Piezo2;hM3Dq mice that developed dyspnea about 20–30 min after CNO injection (i.t., 3 μL of 26 μM CNO solution, $n = 3$), similar to the observations in CNO i.p. treated Piezo2;hM3Dq female mice.

### Intersectional activation of colonic afferent neurons induced colonic hypersensitivity in a sex-dependent manner

To avoid targeting other organs and specifically act on colonic afferent neurons, we applied intersectional strategies. Previous studies showed that optogenetic activation of Piezo2-expressing neurons evoked somatic pain in mice[26]. We therefore implemented optogenetic approaches to restrict the site of activation to the area of thoracolumbar DRG that innervated the distal colon[38]. To do so, we generated mice that contained channelrhodopsin-2 (ChR2) in Piezo2-expressing cells by crossing Ai27D mice with, or by intrathecal injection of adeno-associated viruses pAAV-Ef1a-DIO hChR2 (C128S/D156A)-EYFP (yellow fluorescent protein) into Piezo2-EGFP-IRES-Cre mice. We validated the protocols of using fiber coupled light-emitting diode (LED) @ 470 nm blue light (Thorlabs, Inc.) that activated Piezo2;ChR2-expressing neurons in DRG cell culture. In viral injected Piezo2;ChR2 mice, YFP-ChR2 (Fig. S4a: yellow cells) was co-localized with Piezo2-IR (Fig. S4b: red cells) in DRG neurons (Fig. S4c: orange cells indicated by blue arrows). We loaded DRG neuron culture with a red calcium ($Ca^{2+}$) indicator RhoD4-AM (2 μM; Ex/Em =546/555 nm; Fig. S4d) for optogenetic imaging[39] in which blue light stimulation activated DRG neurons (Fig. S4e, g), KCl stimulation serving as positive control (Fig. S4f, g). We applied the same photostimulation paradigm to DRG explants of Piezo2;ChR2 mice. Blue light stimulations also increased the phosphorylation levels of cAMP response element-binding protein (CREB) in the nucleus of Piezo2;ChR2-containing DRG neurons (Fig. S4h-i), suggesting that this approach activated Piezo2-expressing DRG neurons in whole DRG. Thus, for studies in intact mice we used the same paradigm of blue light stimulations. We implanted the LED-coupled optical fiber in Piezo2;ChR2 mice through an incision toward thoracolumbar DRG areas (Figs. 2a, b). We found that blue light

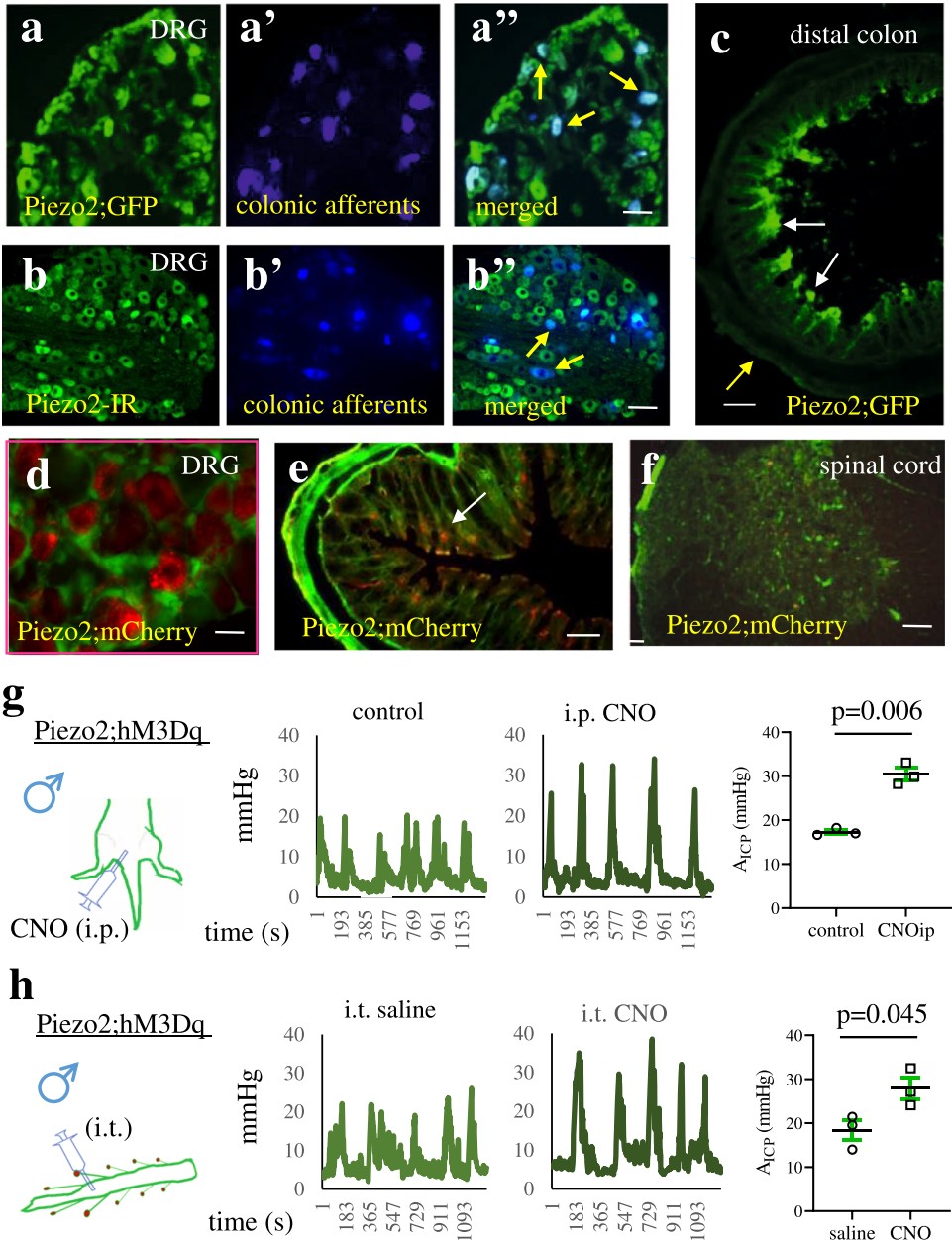

**Fig. 1 | Piezo2 expresses in colonic reflex pathway and activation of Piezo2-expressing cells or Piezo2-expressing sensory neurons elicits colonic hypersensitivity. a-a"** Piezo2;GFP (**a**: green fluorescent protein) mice are injected with Fast Blue (FB) to the muscular wall of the distal colon to specifically label colonic afferent neurons (**a'**: blue cells) in dorsal root ganglia (DRG). Piezo2/GFP expression in colonic afferent neurons is assessed by double imaging for co-localization (**a"**: merged, DRG sections from 3 mice are visualized). Scale bar: 50 μm. **b** − **b"** Wildtype mice are injected with FB to the muscular wall of the distal colon to specifically label colonic afferent neurons in DRG. Piezo2 immunoreactivity (**b**: IR, green cells) is expressed in FB-labeled colonic afferent neurons (**b'**: blue cells) which is demonstrated by double imaging to show co-localization (**b"**). DRG sections from 2 mice are evaluated. Scale bar: 50 μm. **c** Piezo2/GFP expression is examined in transverse sections of the distal colon (2 different segments of colons from 2 mice (a total of 4 segments) are sectioned for visualization). Scale bar: 500 μm. **d** Piezo2;hM3Dq mice show Piezo2/mCherry expression in DRG neurons (3 male and 2 female mice

are evaluated). Scale bar: 20 μm. **e** Piezo2/mCherry expression is examined in transverse sections of the distal colon (3 mice are evaluated). Scale bar: 500 μm. **f** Piezo2/mCherry is not expressed in the spinal cord (2 mice are evaluated). Scale bar: 150 μm. **g** Piezo2;hM3Dq male mice are injected by clozapine N-oxide (CNO, 3 mg/kg body weight. intraperitoneal (i.p.) injection) which elevates the amplitude of intracolonic pressures ($A_{ICP}$), indicating an increase in colonic mechanical sensitivity. $n = 3$ biologically independent animals. Data are presented as mean values +/- SEM. Two-tailed paired $t$ test ($p = 0.0063$, $t = 12.55$, $r = 0.8586$). This dose of CNO injection (i.p.) to the same litter and same age of female Piezo2;hM3Dq mice is lethal. **h** Intrathecal injection (i.t.) of CNO (3 μL of 26 μM CNO solution, $n = 3$ biologically independent animals) to Piezo2;hM3Dq male mice increases colonic mechanical sensitivity when compared to intrathecal injection (i.t.) of saline controls ($n = 3$ biologically independent animals). Data are presented as mean values +/- SEM. Two-tailed unpaired $t$ test ($p = 0.045$, $t = 2.88$, $F = 1.2$). This dose of CNO injection to the same litter and same age of female Piezo2;hM3Dq mice causes dyspnea.

stimulation evoked an elevation in colonic mechanosensing in male mice when compared to male sham that was prior to photostimulation (Fig. 2a). However, localized optogenetic stimulation of thoracolumbar DRG in female Piezo2;ChR2 mice did not change colonic sensitivity (Fig. 2b). Female Piezo2;ChR2 mice also did not show any breathing

problems after localized optogenetic treatment, a stark contrast to CNO treatment of Piezo2;hM3Dq female mice. One of the explanations could be that CNO activated Piezo2-expressing thoracic DRG sensory neurons that innervated the lung. Previous study showed that optogenetic activation of Piezo2-expressing vagal sensory neurons caused apnoea in

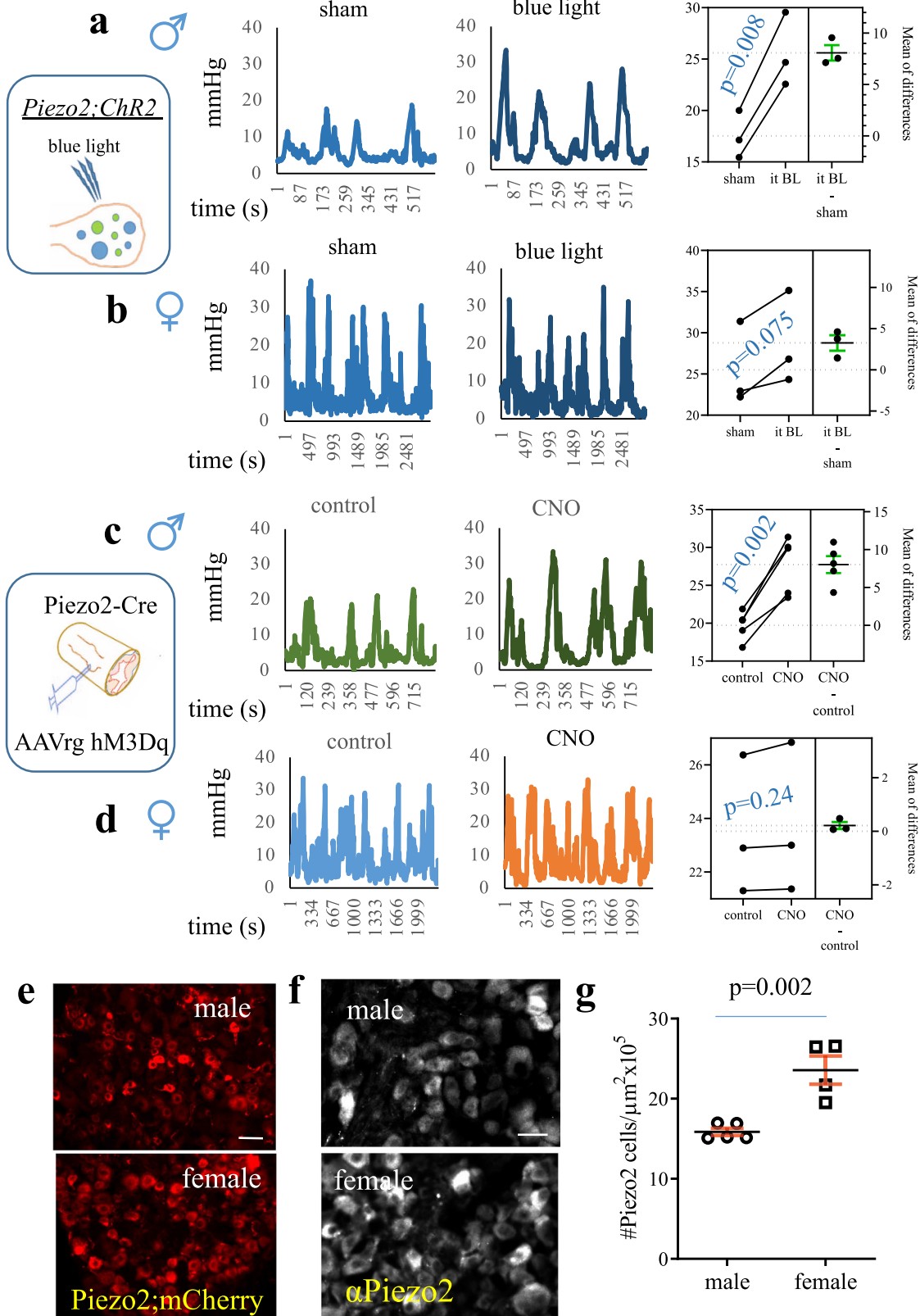

adult mice[23], suggesting that the lung function can be easily disturbed by activation of lung sensory neurons that expressed Piezo2.

We next specifically targeted colonic afferent neurons by injecting retrograde virus pAAV-hSyn-DIO-hM3D(Gq)-mCherry (AAV Retrograde, Addgene ID 44361-AAVrg) into the distal colon of Piezo2-Cre mice (Figs. 2c, d), which was designated as

Piezo2;hM3Dq$^{afferents}$. In these mice, hM3Dq/mCherry was expressed in colonic afferent neurons (Fig. S5a, mCherry was present in T13, L1, L2, L6 DRGs but not in L4 DRGs), therefore intraperitoneal CNO treatment of this intersectional strain allowed specific activation of colonic afferent neurons but not sensory neurons that innervated other organs. We administered CNO to Piezo2;hM3Dq$^{afferents}$ male and

**Fig. 2 | Intersectional activation of Piezo2-expressing colonic afferent neurons evokes colonic hypersensitivity in male mice but not in female mice. a, b** Optogenetic activation of Piezo2-expressing thoracolumbar dorsal root ganglia (DRG) neurons increases colonic mechanical sensitivity in male mice but not in female mice. $n = 3$ biologically independent animals for each group. Data are presented as mean values +/- SEM. Two-tailed paired $t$ test (male: $p = 0.0084$, $t = 10.83$, $r = 0.9973$; female: $p = 0.0751$, $t = 3.441$, $r = 0.9589$). **c, d** Retrograde adeno-associated viruses (AAV) carrying floxed[stop] hM3Dq is injected into the muscular wall of the distal colon of Piezo2-Cre mice to specially label colonic afferent neurons. Clozapine N-oxide (CNO, 2 mg/kg body weight intraperitoneal (i.p.) injection) treatment causes colonic hypersensitivity in male mice ($n = 4$ biologically independent animals). Data are presented as mean values +/- SEM. Two-tailed paired $t$ test: $p = 0.002$, $t = 7.171$, $r = 0.7982$) but not in female mice ($n = 3$ biologically

independent animals. Data are presented as mean values +/- SEM. Two-tailed paired $t$ test: $p = 0.2426$, $t = 1.64$, $r = 0.9998$, no multiple comparisons were made). **e** Representative images of Piezo2 expression demonstrated by Piezo2-Cre driven mCherry products in DRGs of male and female mice. Scale bar: 50 μm. **f** Piezo2-immunoreactivity (IR) in DRG of male and female mice. Scale bar: 50 μm. **g** Summarized data shows the number of Piezo2-expressing neurons in $10^5$ μm$^2$ area of DRG sections of male and female mice. Data are pools from Piezo2;mCherry (male $n = 3$ biologically independent animals, female $n = 2$ biologically independent animals) and Piezo2-IR (male $n = 2$ biologically independent animals, female $n = 2$ biologically independent animals) totaling $n = 5$ biologically independent male mice and $n = 4$ biologically independent female mice. Data are presented as mean values +/- SEM. Two-tailed unpaired $t$ test ($p = 0.0021$, $t = 4.733$, $F = 12.83$).

female mice at a dose of 2 mg/kg body weight. In male Piezo2;-hM3Dq[afferents] mice, CNO treatment induced colonic hypersensitivity (Fig. 2c). In contrast, CNO treatment of Piezo2;hM3Dq[afferents] female mice did not change the mechanosensitivity of the distal colon (Fig. 2d) and also did not cause labored breathing. These observations in CNO-treated Piezo2;hM3Dq[afferents] mice were consistent to the results from localized optogenetic activation of Piezo2-expressing thoracolumbar DRG neurons, suggesting a sex-differential role of Piezo2-expressing sensory neurons in regulation of colonic hypersensitivity.

## Sex differential properties of Piezo2 in sensory neurons

Driven by the sexual dimorphism of Piezo2-expressing sensory neurons in regulation of colonic function, we examined sex-specific nature of Piezo2 expression and function in DRG neurons. We found that the number of Piezo2;mCherry-expressing DRG neurons (Fig. 2e) and the number of Piezo2-IR DRG neurons (Fig. 2f) were higher in female mice than in male mice (Fig. 2g). We next measured the diameter of DRG neurons and surprisingly we found that the sizes of Piezo2-expressing sensory neurons were relatively larger in female mice than in male mice, however, they clustered in small-to-medium (16–36 μm) sized neurons in dehydrated (for cyoprotection) DRG sections (Fig. 3a). We then measured the area of Piezo2-expressing DRG neurons and divided them into small size (100–300 μm$^2$) nociceptors, medium size (300-600 μm$^2$) mechanoreceptors, and large size (>600 μm$^2$) neurons including proprioceptors. In similar total area of DRG sections between male and female mice (12 DRG sections for each gender), female mice had much more Piezo2-expressing DRG neurons with much larger soma size than male mice (Fig. 3b). The cell size distribution of Piezo2-expressing DRG neurons was conserved after birth when comparing between postnatal day (PND)8 and adult male mice (Fig. S5b).

Since Piezo2 distribution profiles in DRG neurons were sex differential in terms of cells sizes, we compared the subtypes of DRG neurons that expressed Piezo2 in male and female mice. Our results showed that the number of Isolectin B4 (IB4)-positive Piezo2-expressng DRG neurons were higher in female mice than in male mice (Figs. 3c–e), suggesting that female mice had more non-peptidergic Piezo2-expressing DRG neurons than male mice. Female and male mice had similar levels of CGRP-positive peptidergic Piezo2-expressing DRG neurons (Fig. 3f–h). Piezo2 also expressed in Nav1.8-expressing polymodal nociceptors (Fig. 4a) and the nociceptors that innervated the distal colon (Fig. S5c). The number of DRG neurons that contained both Piezo2 and Nav1.8 were higher in female mice than in male mice (Fig. 4b), and the percentage of Nav1.8-lineage DRG neurons that expressed Piezo2 was also higher in female mice than in male mice (Fig. 4c). For comparison, we performed Piezo1 immunostaining on different sections from the same sets of DRG. Interestingly, Piezo1 immunoreactivity was located near or on nuclear membrane in some DRG neurons (Fig. S5d) with few in nociceptors (Figs. 4b, c), while in the same sets of DRGs Piezo2 demonstrated cytoplasmic expression (Fig. S5e). To further characterize Piezo1 expression in DRG nociceptors, we performed nociceptor-specific mRNA extraction from Nav1.8;EGFP::L10a mice[40] followed by PCR which showed that the transcripts of Piezo2 (*Fam38b*) but not Piezo1 (*Fam38a*) were found in nociceptive neurons, CGRP (*Calca*) and beta-actin (*ACTB*) serving as control (Fig. 4d). In contrast, Piezo1 was highly expressed in the distal colon and the urinary bladder (Fig. 4d)[41,42], which was consistent to previous studies showing strong Piezo1 expression in enteric neurons and fibers in the colon of guinea, mice and human[30].

We next examined the activity of Piezo2 in DRG nociceptive neurons. Piezo2 channel can be activated by glass pipette poking[17,43]. We loaded cultured DRG neurons with a voltage sensitive dye Di-8-ANEPPS. Poking-evoked changes in the intensity of Di-8-ANEPPS fluorescence in DRG neurons were captured by a charged coupled device (CCD) camera with a frame rate of 1 kHz (1 frame/ms)[44]. We used DRGs from Piezo2;YFG mice to validate this method. Piezo2-expressing DRG neurons in culture were identified by YFP expression (Fig. S6a, green cells). Poking of one of the Piezo2;YFP expressing cells (Fig. S6b) elicited an increase in the intensity of Di-8-ANEPPS fluorescence (compare Fig. S6d to c), which we designated as responders (Fig. S6e). We next applied this technique to nociceptors. We poked nociceptors from Nav1.8;YFG mice (Fig. 4e, green cell). From female mice, 40 out of 48 nociceptors (an average of 83.3%) that we poked were responders, while 23 out of 34 nociceptors (an average of 67.6%) were responders in male mice, a significant lower number when compared to female mice (Fig. 4e). This was consistent to lower levels of Piezo2 expression in nociceptors in male mice than in female mice (Fig. 4b, c). To further assess the role of Piezo2 in nociceptor mechanosensing, we generated Piezo2 conditional knockout (Piezo2[cKO]) mice in which Piezo2 was deleted specifically from nociceptors; Piezo2 intact Nav1.8-Cre$^{+/-}$ mice (Piezo2[wt]) served as genetic control (Fig. S7). We injected adeno-associated viruses pAAV-CAG-FLEX-Archon1-KGC-EGFP-ER2 (Addgene ID 108422-AAV5) into Piezo2[cKO] and Piezo2[wt] mice for Cre-driven labeling of nociceptors with GFP and Archon 1 (Fig. 4f). We performed glass pipette poking of GFP-labeled nociceptive neurons (Fig. 4f, green cells) from Piezo2[wt] and Piezo2[cKO] mice (a representative neuron from Piezo2[wt] mice to demonstrate responders and a representative neuron from Piezo2[cKO] mice to demonstrate nonresponders (Fig. 4f, red cells; Fig. 4f')). In viral injected Piezo2[wt] female mice, 13 out of 16 (81.3%) nociceptive DRG neurons were responders (Fig. 4g), which was equivalent to the data (an average of 83.3%) from Nav1.8;YFP female mice (Fig. 4e). In viral injected Piezo2[cKO] female mice, 3 out of 13 (23.1%) nociceptive DRG neurons responded positively to poking, a marked reduction in the percentage when compared to viral injected Piezo2[wt] or Nav1.8;YFP female mice (Fig. 4g). In viral injected Piezo2[wt] male mice, 8 out of 12 poked nociceptive DRG neurons (66.7%) responded positively (Fig. 4g), which was also equivalent to the data (an average of 67.6%) from Nav1.8;YFP male mice (Fig. 4e). In viral injected Piezo2[cKO] male mice, 3 out of 11 poked nociceptive DRG

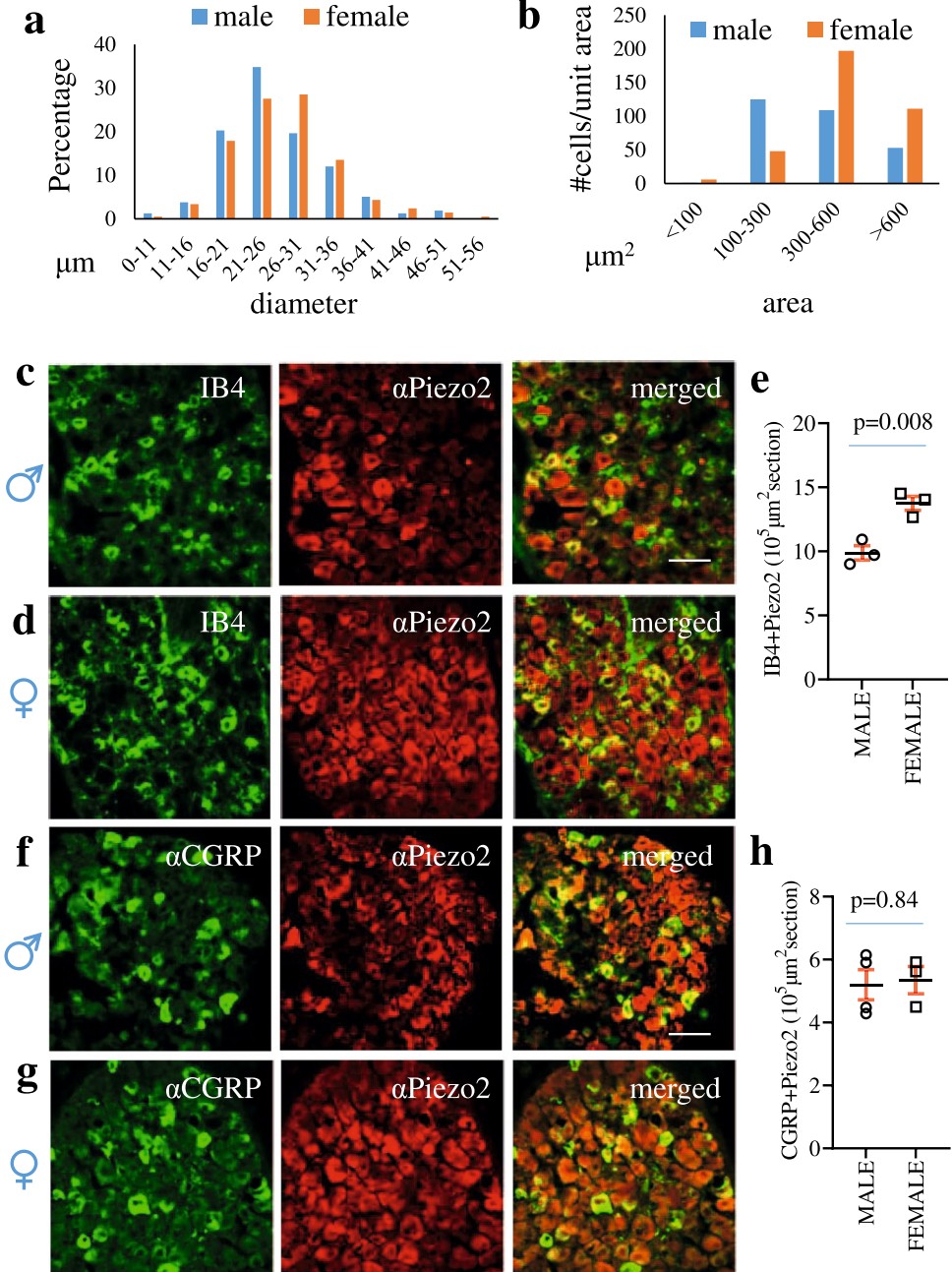

**Fig. 3 | Sex differential expression of Piezo2 in subtypes of DRG neurons. a** The percentage of Piezo2-expressing dorsal root ganglia (DRG) neurons distributes to each category defined by diameters of cell-sizes. Data are from DRG of 3 biologically independent male mice (162 neurons) and DRG of 3 biologically independent female mice (208 neurons). **b** The numbers of Piezo2-expressing DRG neurons are categorized into different groups according to the size of the neurons by area (μm²). Data are from male (318 neurons measured) and female (503 neurons measured) mice. **c**, **d** Double immunostaining of Isolectin-IB4 and Piezo2 in male and female mice. **e** Summarized data shows IB4 and Piezo2 co-expression in 10⁵ μm² neurons. area of DRG sections of $n = 3$ biologically independent male and $n = 3$ biologically independent female mice. Data are presented as mean values +/- SEM. Two-tailed unpaired $t$ test ($p = 0.0079$, $t = 4.92$, $F = 1.041$). Scale bar: 50 μm. **f, g** Double immunostaining of calcitonin gene-related peptide (CGRP) and Piezo2 in male and female mice. **h** Summarized data shows CGRP and Piezo2 co-expression in 10⁵ μm² area of DRG sections of $n = 4$ biologically independent male and $n = 3$ biologically independent female mice. Data are presented as mean values +/- SEM. Two-tailed unpaired $t$ test ($p = 0.84$, $t = 0.2126$, $F = 1.632$). Scale bar: 50 μm.

neurons (27.3%) were responders (Fig. 4g), a reduction when compared to viral injected male Piezo2$^{wt}$ or Nav1.8;YFP male mice.

Osmoregulation occurs constantly in the colon during intestinal fluid balance[45]. Hypoosmolar stimulation (94-194 mosM) has been used to examine the responses of enteric neurons[46]. We therefore examined the responses of nociceptors (Fig. 5a, Fig. S5c: red cells) to hypoosmolar stimulation (nerves of nociceptors can sense changes in intestinal fluid balance) as an alternative mechanical deformation of

DRG neurons using Nav1.8;tdTomato;GCaMP mice. After switching from isotonic solution (289 mosM) to hypoosmolar solution (149 mosM), the sizes of nociceptors were enlarged (Fig. 5a) by an average of 15% (Fig. 5b) with no sex-dependent difference. Hypoosmolar stimulation evoked Ca²⁺ transients in nociceptive neurons measured by increased GCaMP intensity (Figs. 5c, d). The percentage of nociceptive neurons (identified by expression of tdTomato) that responded to hypoosmolar stimulation were higher in female mice than male mice

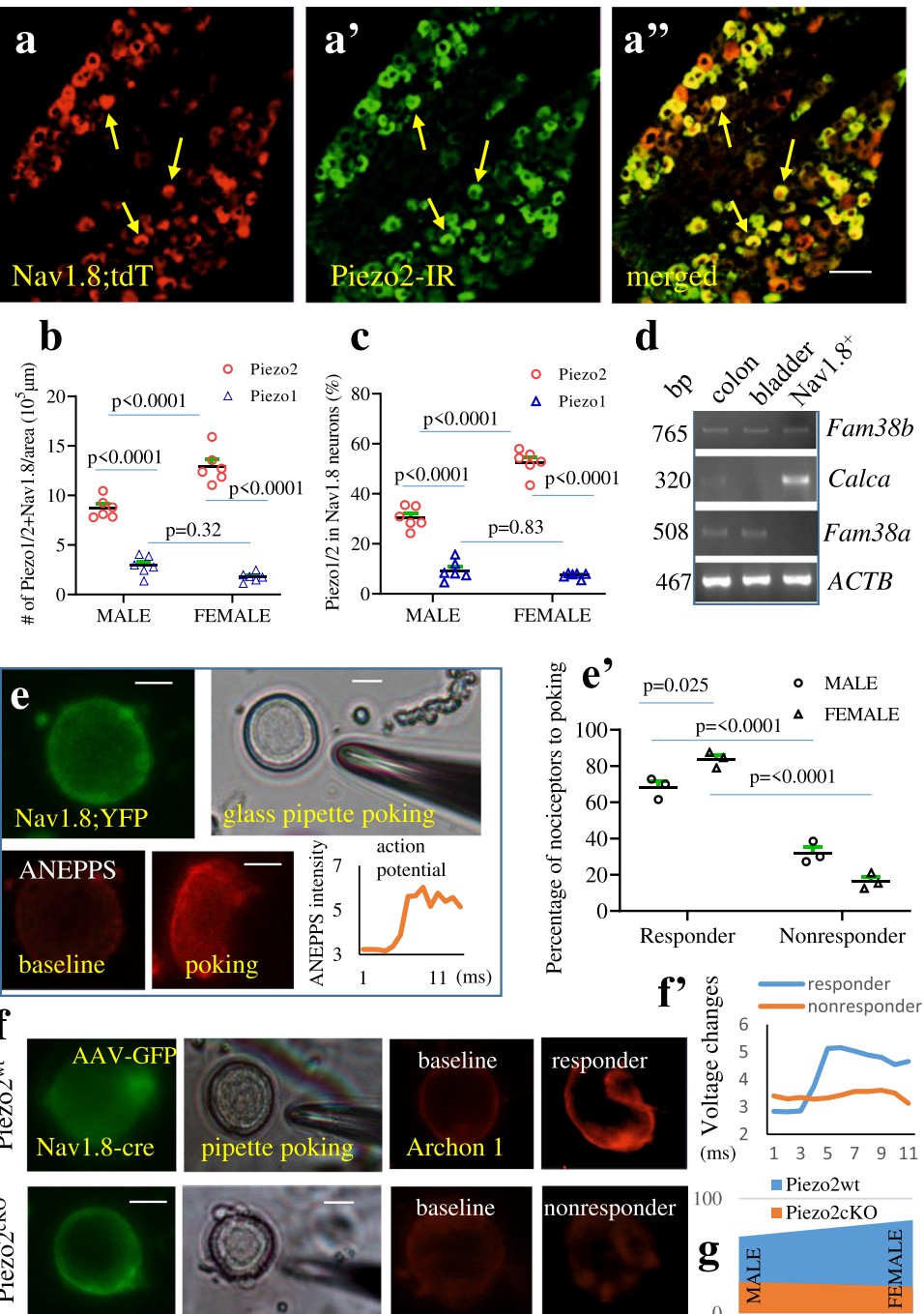

(Fig. 5e). We initially used the conventional inhibitor D-GsMTx4 to block Piezo activity[47] which (20 μM but not 5 μM) reduced $Ca^{2+}$ transients in nociceptors responding to osmoregulation (Fig. 5e). We next injected AAV-EF1a-DIO-GCaMP6s-P2A-nls-dTomato virus (Addgene ID 51082-AAV1) into Piezo2wt and Piezo2cKO mice by which Piezo2 deleted or Piezo2 intact nociceptors were labeled by tdTomato (Fig. 5f, red cells) and also contained GCaMP (Fig. 5f, green cells). Hypoosmolar stimulation evoked $Ca^{2+}$ transients in nociceptors followed by KCl treatment as positive control (Fig. 5g). The numbers of nociceptive neurons that were activated by hypoosmolar stimulation were reduced in Piezo2cKO mice when compared to Piezo2wt mice of both genders (Fig. 5h). Our results from Piezo2 expression in nociceptors (Fig. 4a–d), Piezo2 channel responses to glass pipette poking of nociceptors (Fig. 4e, f), and nociceptive neuron responses to hypoosmolar stimulation in presence or absence of Piezo2 suggested sexual dimorphisms of Piezo2 expression and function in nociceptive neurons.

**Conditional deletion of Piezo2 from nociceptive sensory neurons affected colonic mechanical sensitivity differently by sex**

To assess the role of Piezo2 in nociceptors in regulation of colonic function, we examined the effects of Piezo2 conditional deletion from nociceptive neurons on GI physiology in both genders. We set up the breeding cages at the same time to produce age-matched (week-8) Piezo2wt and Piezo2cKO mice for comparison. Interestingly, we found that Piezo2 deletion affected body weight growth of female mice but not male mice, which were that Piezo2cKO female mice had relatively larger body weight than Piezo2wt female mice at corresponding age stages (Fig. 6a). This suggested that Piezo2 deletion might alter GI function in female mice. Indeed, we found that colonic transit was slower in Piezo2cKO female mice when compared to Piezo2wt female mice (Fig. 6b). We next examined colonic function showing that male Piezo2cKO mice demonstrated similar colonic mechanical sensitivity when compared to male Piezo2wt mice (Fig. 6c), while female Piezo2cKO

**Fig. 4 | The expression and activity of Piezo2 in nociceptive neurons are sexual dimorphic. a–a″** Dorsal root ganglia (DRG) sections containing Nav1.8;tdTomato (**a**: red cells) are immunostained with anti-Piezo2 antibody (**a′**: green cells) to demonstrate expression of Piezo2 in Nav1.8-lineage nociceptive DRG neurons (**a″**: yellow cells show co-localization). Arrows pointed DRG neurons express both Nav1.8 and Piezo2. Scale bar: 50 μm. **b** The numbers of DRG neurons that co-express Piezo2 and Nav1.8 or Piezo1 and Nav1.8 in $10^5$ μm$^2$ area of DRG sections. $n = 6$ DRGs from 3 biologically independent male mice and $n = 6$ DRGs from 3 biologically independent female mice. Data are presented as mean values +/- SEM. Two-way ANOVA with Tukey's multiple comparison test (male Piezo2 vs Piezo1: $p < 0.0001$, $F = 10.99$; male Piezo2 vs female Piezo2: $p < 0.0001$, $F = 337.1$; female Piezo2 vs female Piezo1: $p < 0.0001$, $F = 10.99$; male Piezo1 vs female Piezo1: $p = 0.3248$, $F = 337.1$). **c** The percentage of nociceptive neurons that expressed Piezo2 or Piezo1. $n = 6$ DRGs from 3 biologically independent male mice and $n = 6$ DRGs from 3 biologically independent female mice. Data are presented as mean values +/- SEM. Two-way ANOVA with Tukey's multiple comparison test (male Piezo2 vs Piezo1: $p < 0.0001$, $F = 40.71$; male Piezo2 vs female Piezo2: $p < 0.0001$, $F = 435.4$; female Piezo2 vs female Piezo1: $p < 0.0001$, $F = 40.71$; male Piezo1 vs female Piezo1: $p = 0.826$, $F = 435.4$). **d** Transcriptional analysis of Nav1.8-lineage nociceptive neurons expressing Piezo2 (*Fam38b*), calcitonin gene-related peptide (*Calca*), Piezo1 (*Fam38a*) and beta actin (*ACTB*) using mRNA from the colon and the urinary bladder as control (2 mice are evaluated). **e** Green fluorescent protein (GFP)-labeled nociceptive neurons are loaded with a voltage sensor Di-8-ANEPPS (10 μM) for 20 min and subject to glass pipette poking that elicits voltage changes in a subpopulation of nociceptors in a sex-dependent manner (summarized in **e′**: a total of 34 neurons from $n = 3$ biologically independent male mice and 48 neurons from $n = 3$ biologically independent female mice are poked). Data are presented as mean values +/- SEM. Two-way ANOVA with Tukey's multiple comparison test (male responder vs female responder: $p = 0.0246$, $F = 303.7$; male responder vs male nonresponder: $p = 0.0001$, $F = 0$; male nonresponder vs female nonresponder: $p = 0.0246$, $F = 303.7$; female responder vs female nonresponder: $p < 0.0001$, $F = 0$). Scale bar: 20 μm **f** Adeno-associated viruses (AAV)-GFP/Archon 1-labeled nociceptive neurons from Piezo2$^{wt}$ and Piezo2$^{cKO}$ mice are subject to glass pipette poking. The representative neuron from Piezo2$^{wt}$ mice demonstrates positive response to poking (a responder, the response curve is shown in **f′**). The representative neuron from Piezo2$^{cKO}$ mice demonstrates negative response to poking (a nonresponder, the response curve is shown in **f′**). Scale bar: 20 μm **g** Summarized data shows the percentage of nociceptive neurons that respond to poking (the number of biologically independent neurons poked are $n = 16$ from Piezo2$^{wt}$ female, $n = 13$ from Piezo2$^{cKO}$ female, $n = 12$ from Piezo2$^{wt}$ male, and $n = 11$ from Piezo2$^{cKO}$ male mice).

mice had reduced colonic mechanical sensitivity when compared to female Piezo2$^{wt}$ mice (Fig. 6d). The expression level of CGRP in thoracolumbar DRG stayed the same for male Piezo2$^{wt}$ and Piezo2$^{cKO}$ mice (Fig. 6e) but was dramatically reduced in female Piezo2$^{cKO}$ mice when compared to female Piezo2$^{wt}$ mice (Fig. 6f).

It is noteworthy that female Piezo2$^{wt}$ mice had greater basal colonic mechanical sensitivity than male Piezo2$^{wt}$ mice when the same conditions were applied (Figs. 6c, d: female $A_{ICP} = 23.85 ± 1.12$ mmHg; male $A_{ICP} = 19.5 ± 1.15$ mmHg; Unpaired $t$-test: $p = 0.035$). In addition to the sex differential expression and activity of Piezo2 in nociceptors (Figs. 3–5) as one of the possible underlying mechanisms, some studies also suggested a sex-dependent involvement of immune cells in mechanical pain hypersensitivity[48], especially M1-like microphages being reported to participate, independent of Nav1.8 nociceptors, in the maintenance of osteoarthritis pain where gender was not specified[49]. Therefore, we examined the levels of CD80$^+$ M1-like macrophages in DRG of Piezo2$^{wt}$ male and female mice (Fig. S8a). We showed that there was no significant sex-dependent difference (Fig. S8b-c). Thus, macrophages were not associated to the sexual difference in the baseline of colonic mechanical sensitivity.

## Piezo2 expressed in nociceptive neurons participated in colonic mechanical pain hypersensitivity in male mice but not in female mice

Colonic inflammation induced by 2,4,6-trinitrobenzene sulfonic acid (TNBS) led to colonic hypersensitivity[36] and Piezo2 up-regulation in DRG[29]. Reduction of Piezo2 levels in DRG by intrathecal Piezo2-short hairpin RNA (shRNA) attenuated painful behavioral responses to colorectal distension (CRD) in wildtype rats[29]. Since Piezo2 conditional deletion from nociceptors did not affect colonic mechanical sensitivity in healthy male mice (Fig. 6), we measured colonic mechanical sensitivity in male Piezo2$^{wt}$ and Piezo2$^{cKO}$ mice following induction of colonic inflammation. We found that colonic inflammation enhanced colonic mechanical sensitivity in Piezo2$^{wt}$ mice (Fig. 7a). However, colonic inflammation in Piezo2$^{cKO}$ mice did not change colonic mechanical sensitivity when compared to uninflamed Piezo2$^{cKO}$ mice (Fig. 7a; summary in Fig. 7b). The CRD threshold to elicit painful behavior in inflamed mice was significantly smaller in Piezo2$^{wt}$ mice than in Piezo2$^{cKO}$ mice (Fig. 7c). These findings suggested that Piezo2 nociceptors mediated the development of colonic hypersensitivity caused by colonic inflammation in male mice. Interestingly, colonic inflammation in male mice (on day 3) failed to increase (in contrast, decreased) macrophage levels in DRG when identified by macrophage colony-stimulating factor 1 receptor (Csf1R)-GFP (Fig. S8d). This again suggests that Piezo2 in nociceptive neurons participated in the regulation of colonic mechanical sensitivity independent of macrophages.

We next examined the role of Piezo2 in nociceptive neurons in mediating noxious colonic stimulation in male mice. We applied CRD at 80 mmHg 3 times with 10-s durations and 5-s intervals as a noxious mechanical force to the colon wall as we and others used previously[36,50]. We further validated this noxious CRD paradigm using in vivo Ca$^{2+}$ imaging of whole L2 DRG which evoked Ca$^{2+}$ transients detected by GCaMP (Fig. S9a-c) but did not traumatize colonic mechanical responses or cause de-sensitization of DRG neurons (Fig. S9d). Therefore, we applied this noxious CRD paradigm to Piezo2$^{wt}$ and Piezo2$^{cKO}$ male mice. In Piezo2$^{wt}$ mice, noxious CRD stimulation reduced CGRP immunoreactivity examined in thoracolumbar DRGs 30 min after CRD was applied (Fig. 7d). However, the same paradigm of noxious CRD did not reduce CGRP expression in Piezo2$^{cKO}$ mice (Fig. 7e). In the corresponding spinal cord of Piezo2$^{wt}$ mice, noxious CRD stimulation led to extensive CGRP release into the spinal cord at the region of dorsal horn (DH) which also extended to the central commissure (CC), while noxious CRD stimulation did not cause much CGRP central release in Piezo2$^{cKO}$ mice (Fig. 7f). These results implicated that Piezo2 in CGRP-expressing cells had a role to mediate CGRP central release, a critical step in spinal central sensitization, following noxious CRD stimulation. In Piezo2$^{cKO}$ male mice, a lack of Piezo2 prevented CGRP central release following noxious CRD stimulation.

We previously reported that the level of p-Akt was up-regulated in thoracolumbar DRG by noxious CRD stimulation of male mice[36]. Here we showed that Piezo2 conditional deletion significantly reduced CRD-evoked p-Akt in thoracolumbar DRG of male mice (Fig. S10a). The activity of CREB is often regulated by Ca$^{2+}$ activity in cells. We found that noxious CRD stimulation induced less CREB activity (phosphorylation level) in thoracolumbar DRG of Piezo2$^{cKO}$ male mice when compared to Piezo2$^{wt}$ male mice (Fig. S10b). Taken together, Piezo2 deletion (less Ca$^{2+}$ channel activity) reduced CREB and Akt activity in DRG neurons and reduced CGRP central release to cause spinal central sensitization in noxious CRD-induced colonic pain responses.

Although Piezo2 conditional deletion reduced the basal level of colonic mechanical sensitivity in female Piezo2$^{cKO}$ mice when compared to female Piezo2$^{wt}$ mice (Fig. 6d), we were curious about how Piezo2 deletion would affect the development of colonic hypersensitivity in female mice. We induced colonic inflammation to female mice using the same protocol as to male mice. In female Piezo2$^{wt}$ mice, colonic inflammation caused a modest increase in colonic mechanical

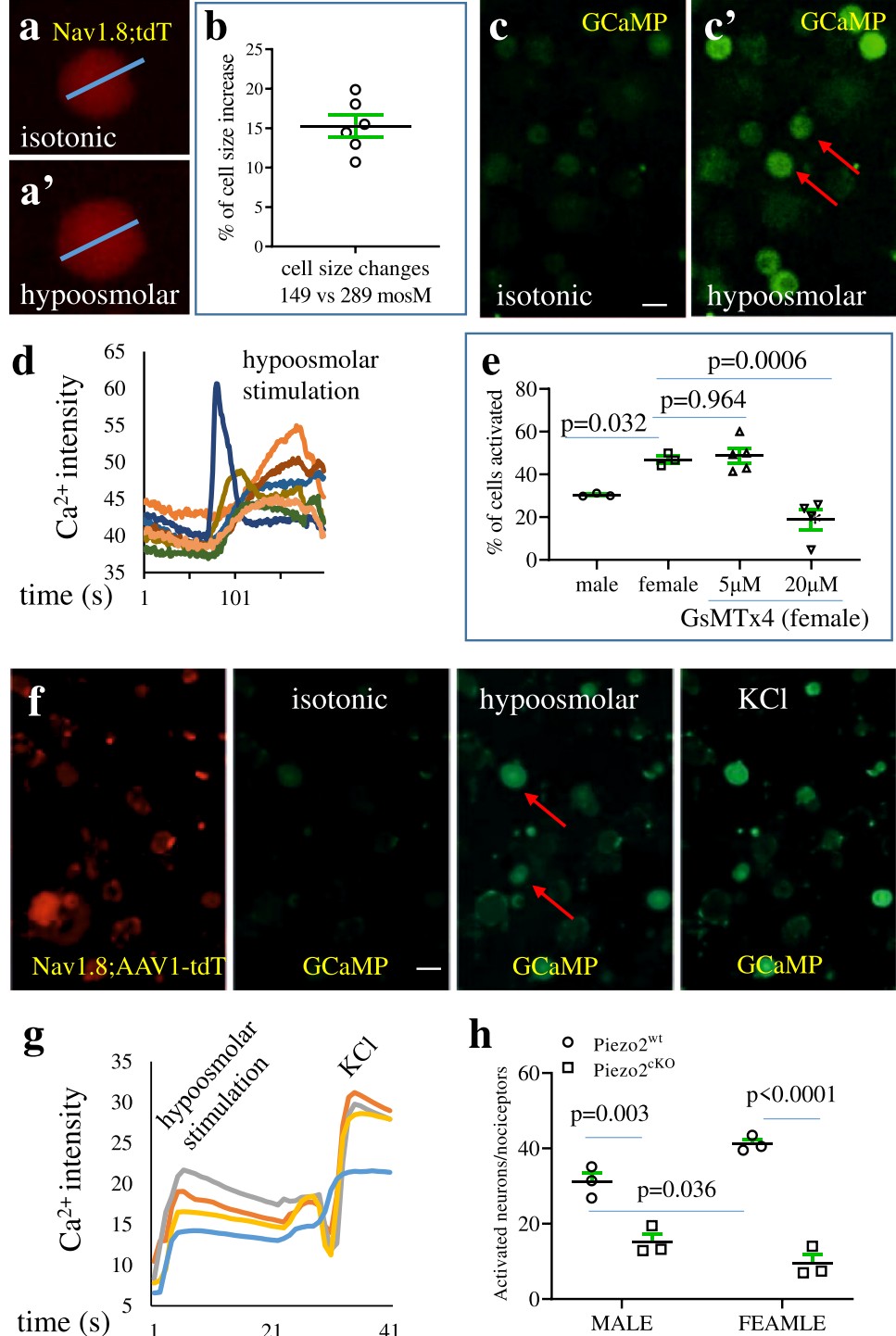

sensitivity (Fig. 8a). Surprisingly, colonic inflammation robustly increased colonic mechanical sensitivity in female Piezo2[cKO] mice when compared to control female Piezo2[cKO] mice (Fig. 8b, summarized in Fig. 8c), suggesting that conditional deletion of Piezo2 from female mice did not suppress the development of colonic pain hypersensitivity. This was in contrast to what was seen in male mice (Fig. 7).

Finally, we analyzed our measurements of somatic mechanical sensitivity of Piezo2[wt] and Piezo2[cKO] mice in response to von Frey filament stimulation of the plantar surface of hindpaw (Figs. 8c, d). This measurement was to supplement our findings in alterations of colonic mechanical sensitivity and hypersensitivity due to Piezo2 targeted deletion. In response to smaller size of von Frey filaments (male <0.16 g; female <0.4 g) that did not evoke pain hypersensitivity in any

of the mice examined, female Piezo2[cKO] mice developed mechanical hyposensitivity when compared to female Piezo2[wt] mice (Fig. 8c), in contrast, male Piezo2[wt] and Piezo2[cKO] mice demonstrated less differential hindpaw withdrawal responses (Fig. 8d). In response to larger size of von Frey filament stimulation that caused all Piezo2[wt] mice (100%) to experience pain (male at 0.4 g, female at 0.6 g), 80% of female Piezo2[cKO] mice and 50% of male Piezo2[cKO] mice demonstrated painful responses. These data suggested that Piezo2 conditional deletion reduced noxious mechanical sensitivity more dramatically in male mice than in female mice. Taken together with colonic mechanical sensitivity and hypersensitivity measurements, our results inferred that Piezo2 in Nav1.8[+] sensory neurons regulated innocuous mechanical sensitivity more prominently in female mice than in male mice and

**Fig. 5 | Piezo2 mediates the responses of nociceptive neurons to hypoosmolar stimulation. a, a'** Changes in the size of nociceptive neurons in the process of switching incubation solution from isotonic condition (**a**) to hypoosmolar condition (**a'**). **b** The percentage of diameter enlargement of nociceptive neurons in the process of switching incubation solution from isotonic condition to hypoosmolar condition. $n = 6$ biologically independent wells (32 neurons were randomly selected and measured). Data are presented as mean values +/- SEM. **c, c'** Changes in the GCaMP intensity in nociceptive neurons (c: GCaMP baseline) upon application of hypoosmolar solution **c'**: green cells show GCaMP expression). Scale bar: 50 µm. **d** Representative calcium ($Ca^{2+}$) transients measured by intensity of GCaMP in nociceptive neurons. **e** The percentage of nociceptive neurons that is elicited for $Ca^{2+}$ activity upon application of hypoosmolar solution with or without D-GsMTx4. $n = 3$ biologically independent wells from male mice, $n = 3$ biologically independent wells from female mice, $n = 5$ biologically independent wells from female mice plus

5 µM D-GsMTx4, and $n = 4$ biologically independent wells from female mice plus 20 µM D-GsMTx4. Data are presented as mean values +/- SEM. One-way ANOVA with Dunnett's Multiple Comparison Test against female mice (male vs female, $p = 0.032$; female vs female 5 µM D-GsMTx4: $p = 0.9639$; female vs female 20 µM D-GsMTx4: $p = 0.0006$; $F = 17.21$). **f** AAV-tdTomato/GCaMP-labeled nociceptive neurons from Piezo2$^{wt}$ and Piezo2$^{cKO}$ mice are subject to hypoosmolar stimulation. Scale bar: 50 µm. **g** Changes in the GCaMP intensity in nociceptive neurons upon application of hypoosmolar solution. **h** The percentage of nociceptive neurons from Piezo2$^{wt}$ and Piezo2$^{cKO}$ mice responds to hypoosmolar stimulation. $n = 3$ biologically independent animals for each group. Data are presented as mean values +/- SEM. Two-way ANOVA with Tukey's multiple comparison test (male Piezo2$^{wt}$ vs male Piezo2$^{cKO}$: $p = 0.0026$, $F = 1.103$; male Piezo2$^{wt}$ vs female Piezo2$^{wt}$: $p = 0.0355$, $F = 133.8$; female Piezo2$^{wt}$ vs female Piezo2$^{cKO}$: $p < 0.0001$, $F = 1.103$).

mediated mechanical pain hypersensitivity more prominent in male mice than in female mice.

## Discussion

The current study reveals a previously uncharacterized phenomenon that Piezo2 expressed in nociceptive neurons demonstrates sexual dimorphism in the mediation of colonic mechanosensing. We find that Piezo2 is expressed in colonic afferent neurons of DRG. Chemogenetic activation of Pizeo2-expressing cells or DRG neurons reveals differential outcomes in male and female mice in inducing colonic hypersensitivity. The levels of Piezo2 expression and activity in DRG are higher in female mice than in male mice. Conditional deletion of Piezo2 from Nav1.8 nociceptive neurons increases body weight growth, slows colonic transits, and reduces colonic mechanical sensitivity in female mice but not in male mice. However, Piezo2 in nociceptive neurons participates in the development of colonic hypersensitivity and pain sensing in male mice but not in female mice. These results suggest that Piezo2 is critical in maintenance of colonic mechanosensory homeostasis in female mice and has an essential role in colonic hypersensitivity in male mice.

It is perplexing for the underlying mechanisms by which female Piezo2;hM3Dq mice are much more sensitive than male Piezo2;hM3Dq mice to systemic CNO treatment that targets all levels of sensory neurons (Fig. 1). In addition, intersectional activation of Piezo2-expressing colonic afferent neurons causes colonic hypersensitivity in male mice but not in female mice (Fig. 2). More surprisingly, conditional deletion of Piezo2 from nociceptive neurons causes bodyweight overgrowth and hyposensitivity in innocuous mechanosensing of the hindpaw and colon of female mice but not male mice (Figs. 6 and 8). To search for the underlying explanations, we find that the expression and activity of Piezo2 in DRG neurons are distinct in male and female mice, showing that female mice possess higher levels of Piezo2 and nonpeptidergic Piezo2-expressing DRG neurons than male mice (Figs. 2 and 3). Transcriptomic analysis reveals consistent results that *Fam38b* counts are 13% less in DRG neurons of male mice than female mice[51]. We reason that the higher level of endogenous Piezo2 in DRG of female mice entitles its prominent function in mechanosensing in which activation or deletion of Piezo2 from DRG of female mice may disturb its homeostatic role in physiology. While, in male mice the lower level of Piezo2 may not make a noticeable difference or other compensatory pathways may exist, for example, the expression of mechanosensitive channels TrpC1 and TrpA1 are much higher in DRG of male mice than DRG of female mice[51]. Piezo2 in colonic epithelial cells[8,33–35] and Piezo channels in enteric neurons[30] may also compensate colonic mechanical sensitivity in Piezo2$^{cKO}$ male mice. The sex-differential role of Piezo2 in human sensory impairments has also been suggested in case studies. Summarized by Yamaguchi et al. in Table 2[52], among the 4/7 patients who possess loss-of-function Piezo2 and experience impaired sensation, 3 out of the 4 are females. Peripheral sensory nerve conduction studies show decreased amplitudes

of sensory nerve action potentials in patient[52], indicating sensory axonal neuropathy in Piezo2 deficient subjects. This discovery of the sexual dimorphism of Piezo2 may shed light on future endeavors in understanding the in-depth molecular mechanisms underlying sex-differential mechanosensory responses.

The involvement of Piezo2 in mechanical pain is reported in a number of studies. Most of these studies have examined somatic pain hypersensitivity[11,21,26,28,53], with a recent publication on the role of Piezo2 in TrpV1-lineage neurons in mediation of visceral mechanical hypersensitivity[54]. Since mice with constitutive ablation of Piezo2 from DRG do not survive beyond 24 hours (h) of birth[23], the approaches of conditional deletion of Piezo2 from subpopulations of DRG neurons have been utilized. For example, mice with Piezo2 deletion driven by HoxB8-Cre that induces a robust Piezo2 deletion from DRG neurons have impaired nocifensive responses to mechanical stimulation of the hindpaw[26]. Mice that have incomplete Piezo2 deletion driven by Advil-creERT2 demonstrate reduced somatic sensitivity to light-touch stimuli of the paws and cornea[24] but not to noxious mechanical stimuli[11,26]. In contrast, Piezo2 deletion driven by Advil-Cre1 that deletes Piezo2 from a portion of LTMRs sensitizes acute hindpaw mechanical pain responses[53]. These studies suggest that the approaches used to genetically delete Piezo2 and Piezo2 expressed in different cell types[18] are critical in behavioral outcomes. In the current study, we delete Piezo2 from Nav1.8-expresssing nociceptors due to the following reasons (1) Nav1.8 nociceptors are polymodal and mediate a variety of pain modalities, including colonic hypersensitivity[36,55], (2) Piezo2 is largely expressed by small-to-medium sized DRG neurons and co-expressed with Nav1.8 in DRG (Fig. 4), and (3) the expression and activity of Piezo2 in Nav1.8-expressing DRG neurons is sex differential (Figs. 4 and 5). We find that Piezo2 deletion increases female body weight growth with slower colonic transits, reduced colonic mechanical sensitivity, and reduction of CGRP expression in DRG neurons (Fig. 6). CGRP is an excitatory neurotransmitter and its level is positively correlated to colonic hypersensitivity and spinal central sensitization[38,56]. Reduction of colonic transits and tactile sensitivity in aging populations are also related to loss of Piezo2 in EC cells[57]. In colonic inflammation-induced colonic hypersensitivity, Piezo2 deletion suppresses the development of colonic mechanical pain in male mice but not in female mice (Figs. 7 and 8). These observations are consistent to that Piezo2 has higher expression levels in medium-sized mechanoceptors of female mice in which when Piezo2 is removed it could sensitize mechanical pain[53]. Measurement of somatic mechanical sensitivity in response to von Frey filament stimulations in Piezo2$^{wt}$ and Piezo2$^{cKO}$ mice also reveals a sexual difference, which is that Piezo2 engages more in innocuous mechanical stimulations in female mice and noxious mechanical stimulations in male mice (Fig. 8). Of noting, intersectional activation of Piezo2-expresing colonic afferent neurons or colonic inflammation can markedly induce colonic hypersensitivity in male mice (Figs. 2 and 7), but fails to or very mildly (1.13-fold in inflammation) induces colonic hypersensitivity in female mice (Figs. 2

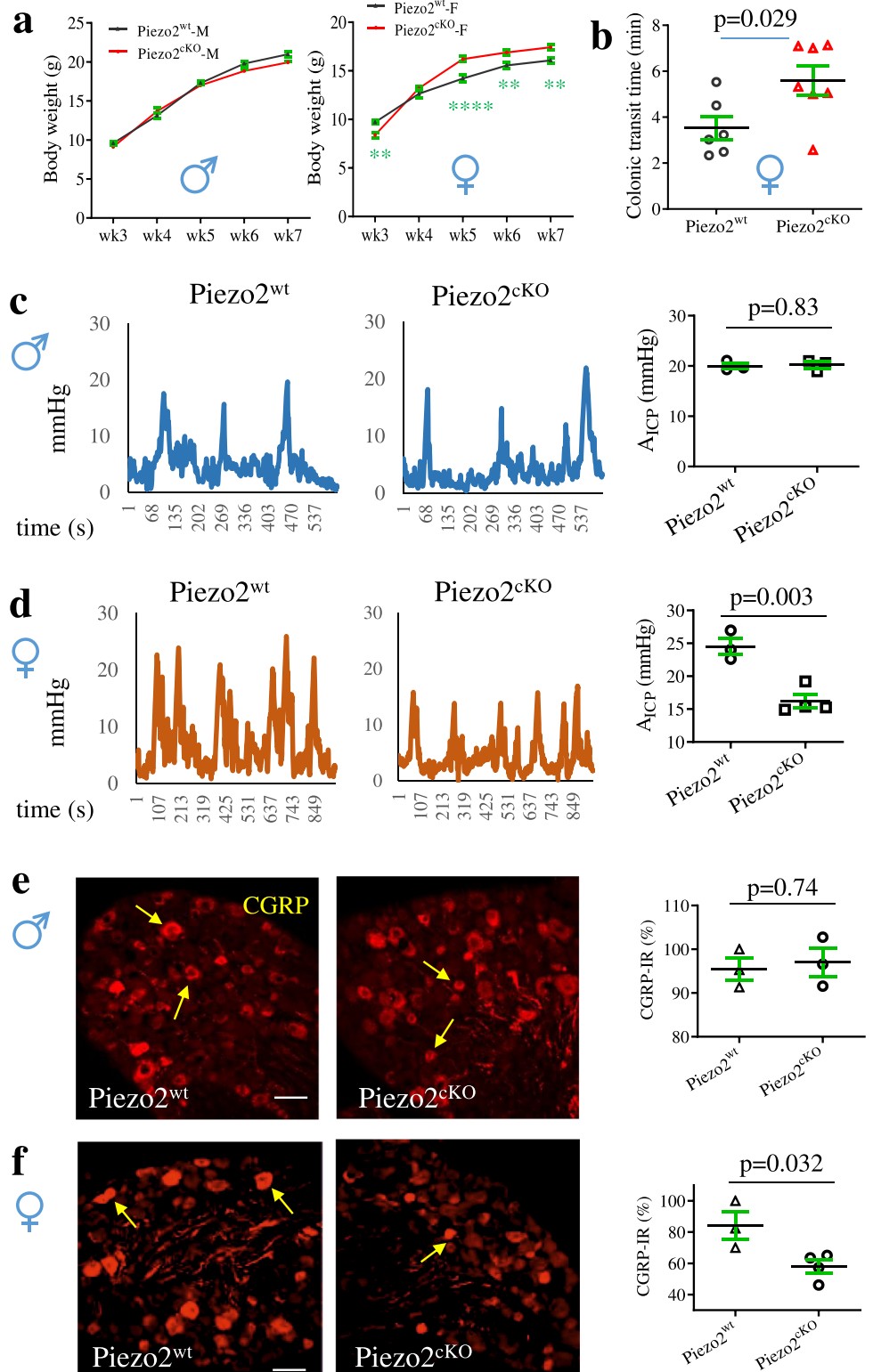

and 8). Suggested by previous studies[53] and our current cell-type specific characterization of Piezo2-expressing DRG neurons (Fig. 3), it is possible that subtypes of Piezo2-expressing DRG neurons may have differential roles in mediating colonic mechanosensing and hypersensitivity, which necessitates further investigations.

In summary, the present study demonstrates an undescribed phenomenon that Piezo2 acts in a sex-specific manner in colonic mechanosensing and pain hypersensitivity. Men and women experience pain differently. This could open a door for future research

endeavors to study the underlying mechanisms of Piezo2 in sexual dimorphism of visceral hypersensitivity and mechanical pain at large.

## Methods
### Animals and breeding strategies
The mice used in this study were as following: Piezo2;GFP mice that were generated by crossing Piezo2-EGFP-IRES-Cre mice (Piezo2-Cre, JAX Stock # 027719, B6(SJL))[14] with EGFP::L10a mice (JAX Stock # 024750, B6); Piezo2;hM3Dq mice that were generated by crossing

**Fig. 6 | Conditional deletion of Piezo2 from nociceptive neurons affects colonic mechanical sensitivity in a sex-dependent manner. a** Mice of Piezo2[wt] (male $n = 13$, female n = 10 from 3 litters) and Piezo2[cKO] (biologically independent male mice $n = 17$, biologically independent female mice $n = 12$ from 4 litters) are weighted on day 21 (wk3), day 28 (wk4), day 35 (wk5), day 42 (wk6), and day 49 (wk7). Data are presented as mean values +/- SEM. Two-way ANOVA with Šídák's multiple comparisons (Male Piezo2[wt] vs Piezo2[cKO]: wk3, $p = 0.7955$, $t = 1.103$; wk4, $p = 0.6418$, $t = 1.33$; wk5, $p = 0.9623$, $t = 0.7069$; wk6, $p = 0.1245$, t = 2.246; wk7, $p = 0.0742$, $t = 2.456$. Female Piezo2[wt] vs Piezo2[cKO]: wk3, $p = 0.008$ (**), $t = 3.242$; wk4, $p = 0.5884$, $t = 1.407$; wk5, $p < 0.0001$ (****), t = 4.966; wk6, $p = 0.0063$ (**), $t = 3.316$; wk7, $p = 0.0056$ (**), t = 3.353). **b** Comparison of colonic transit between Piezo2[wt] ($n = 6$) and Piezo2[cKO] ($n = 7$) biologically independent female mice. Data are presented as mean values +/- SEM. Two-tailed unpaired $t$ test ($p = 0.0289$, $t = 2.511$, $F = 1.755$). **c** Colonometrical recording of male Piezo2[wt] ($n = 3$) and Piezo2[cKO] ($n = 3$) biologically independent mice. The amplitudes of intracolonic pressures ($A_{ICP}$) are characterized. Data are presented as mean values +/- SEM. Two-tailed unpaired $t$ test ($p = 0.8294$, $t = 0.23$, $F = 1.399$). **d** Colonometrical recording of female Piezo2[wt] ($n = 3$) and Piezo2[cKO] ($n = 4$) biologically independent mice. Data are presented as mean values +/- SEM. Two-tailed unpaired $t$ test ($p = 0.0034$, $t = 5.209$, $F = 1.191$). **e** Calcitonin gene-related peptide (CGRP) immunoreactivity in male Piezo2[wt] ($n = 3$ biologically independent mice with 2 dorsal root ganglia (DRG) from each animal totaling 29 sections) and Piezo2[cKO] ($n = 3$ biologically independent mice with 2 DRGs from each animal totaling 28 sections) mice. Data are presented as mean values +/- SEM. Two-tailed unpaired $t$ test (p = 0.741, t = 0.3543, F = 1.676). Scale bar: 50 µm. **f** CGRP immunoreactivity in female Piezo2[wt] ($n = 3$ biologically independent mice with 2 DRGs from each animal totaling 24 sections) and Piezo2[cKO] ($n = 4$ biologically independent mice with 2 DRGs from each animal totaling 26 sections) mice. Data are presented as mean values +/- SEM. Two-tailed unpaired $t$ test ($p = 0.0323$, $F = 3.079$). Scale bar: 50 µm.

Piezo2-Cre mice with RC::L-hM3Dq mice (JAX Stock # 026943, B6); Wt;hM3Dq mice that were generated by crossing wildtype mice (C57BL/6) with RC::L-hM3Dq mice; Piezo2;ChR2 mice that were generated by either crossing Ai27D mice (JAX Stock # 012567, B6) or Ai32 mice (JAX Stock # 012569, B6) with or by intrathecal injection of pAAV-Ef1a-DIO hChR2 (C128S/D156A)-EYFP virus (Addgene # 35503) into Piezo2-Cre mice; Nav1.8;tdTomato;GCaMP mice that were generated by crossing Nav1.8-Cre[+/+] mice (a line that was created by Dr. John Wood, Wolfson Inst. UK)[58] with PC-G5-tdT mice (JAX Stock # 024477, B6); Nav1.8;EGFP::L10a mice that were generated by crossing Nav1.8-Cre mice with EGFP::L10a mice; Piezo2;YFP mice and Nav1.8;YFP mice that were generated by crossing Ai32 mice with Piezo2-Cre or Nav1.9-Cre mice, respectively; Piezo2[cKO] mice that were generated by using Nav1.8-Cre[+/+] mice and floxed Piezo2 (Piezo2[fl/fl]) mice (JAX Stock # 027720, B6(SJL)) which was selected each step for needed genotypes; Piezo2[wt] mice that were generated by crossing Nav1.8-Cre[+/+] mice with wildtype mice; Csf1R;GFP mice that were generated to genetically label macrophages with GFP, which was produced by breeding Csf1r-Cre mice (JAX Stock # 029206, C57BL/6) with EGFP::L10a mice. The sequences of primers for genotyping Nav1.8-Cre (13Salt and Cre 5a) and wildtype (13Salt and 12 A) alleles were: 13Salt: GGAATGGGATG-GAGCTTCTTAC; 12 A: TTACCCGGTGTGTGCTGTAGAAAG; CRE 5a: CAAATGTTGCTGGATAGTT TTTACTGCC. The rest of the genotyping utilized sequences of primers provided by The Jackson Laboratory. All our primers used for genotyping and other PCR were synthesized by Integrated DNA Technologies, Inc. (IDT: Coralville, Iowa).

Male and female mice that were used to generate data were at 2–3-month old except being specifically indicated in each experiment. The parental mice were either purchased from The Jackson Laboratory (JAX, stain numbers were listed above) or obtained from other investigators. Standard husbandry conditions with 12:12-h light cycles and free access to regular food/water were provided to each cage that housed 2–5 mice to ensure adequate social environment. All experimental protocols involving animal use were approved by Virginia Commonwealth University Institutional Animal Care and Use Committee (IACUC). Animal care was in accordance with the Association for Assessment and Accreditation of Laboratory Animal Care (AAALAC) guidelines.

### Retrograde labeling of colonic afferent neurons with neuronal tracing dye or retrograde virus
We injected neuronal tracing dye Fast Blue (FB) into the wall of the distal colon to specifically label colonic afferent neurons in DRG[59]. Specifically, the mouse distal colon was exposed with a lower abdominal incision under anesthesia (2.5% isoflurane) and in a sterile environment. A total of 20 µL of FB (4%, weight/volume; Polysciences, Inc. Warrington, PA) was injected at 4 random spots within a one-cm length of the colon segment, which was 3 cm proximal to the anus. The retrograde AAV virus (Addgene ID 44361-AAVrg) was injected in the same manner. A total of 5 µL of originally packed virus was diluted into 20 µL saline for injection. A Hamilton syringe was used for injection of FB or virus. Injections into the lumen, major blood vessels, or overlying fascial layers were avoided. After the incision was closed by a 4-0 suture, animals were allowed for survival for 5 days to ensure adequate time for FB transport to DRG. Virus-injected mice were used 2 months after injection to ensure proper labeling of colonic afferent neurons.

### Immunostaining and DRG neuron analysis
After fixation with 4% paraformaldehyde, DRGs and the spinal cord were incubated in 25% sucrose overnight at 4 °C for cryoprotection. DRGs were sectioned at 10 µm thickness and the spinal cord at 20 µm thickness. The tissue sections were processed for on-slide immunostaining. The primary antibodies were diluted in PBST (0.3% Triton X-100 in 0.1 M PBS, pH 7.4) buffer containing 5% normal donkey serum (Jackson ImmunoResearch, West Grove, PA) and applied to tissue sections and incubated overnight at room temperature, followed by fluorescence-conjugated species-specific secondary antibody (1:500, Molecular Probes, Eugene, OR) incubation at room temperature for 2 h. The primary antibodies used were rabbit anti-Piezo2 (1:500, Novus Biologicals LLC, Cat# NBP1-78624), rabbit anti-CGRP (1:1000, Invitrogen, Cat# PA5-114929), goat anti-CGRP (1:2000, Abcam, Cat# AB36001), goat anti-Sub P (1:500, Santa Cruz, Cat# sc-9758), rabbit anti-PGP9.5 (1:1000, Millipore, Cat# AB1761-I), rabbit anti-Piezo1 (1:400, Novus Biologicals LLC, Cat# NBP1-78537), rabbit anti-p-CREB (1:500, Cell Signaling, Cat# 9198), and rabbit anti-p-Akt (1:500, Cell Signaling, Cat# 4060). Isolectin GS-IB4 was from Molecular Probes (1:500, Cat# I21411). The specificity of primary antibodies was evaluated by either western blot or pre-absorption assay by us, other publications, or the manufactures. Immunostaining in the absence of primary or secondary antibody was assessed for background evaluation. The secondary antibodies used were donkey anti Rabbit (Cy3) (1:500, Jackson Immuno research, Cat# 711-165-152), donkey anti-goat 594 (1:500, Life Technologies, Cat# A11058), donkey anti-rabbit 488 (1:500, Life Technologies, A21206), donkey anti goat 488 (1:500, Life Technologies, Cat# A11055). After coverslip with Citifluor (Citifluor Ltd., London) mounting medium, tissue sections were viewed and analyzed using a Zeiss fluorescent microscope.

The neurons that showed visible nucleus were measured and counted with a built-in Zeiss software in a blind fashion to minimize any potential bias. For counting, data from multiple sections of a given DRG were pooled and averaged as one data point (raw data were included in the source data). The areas that contained neuronal soma, avoiding the areas that had extensive nerve fibers, were measured for normalizing the expression levels of protein of interest, which was presented as number of positive neurons per unit area. For diameter measurement, all positive neurons from each section were measured and all measurements were pooled to generate the distribution graph. To avoid double counting, we chose every third section in the serial cutting for each specific antibody analysis.

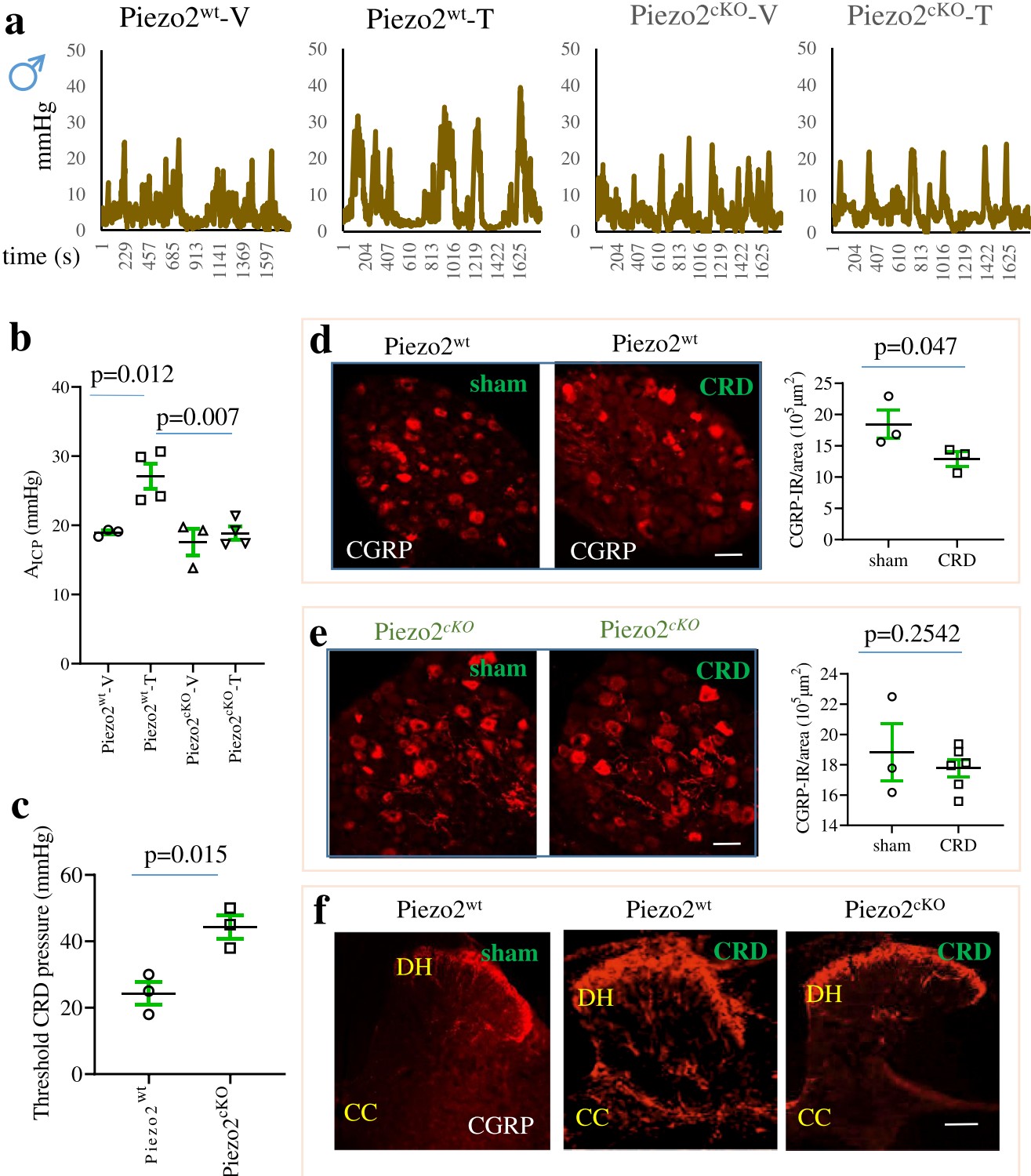

## Colonometry and colonometrogram analysis

To perform colonometry by which the colonic mechanosensitivity was quantitatively and objectively characterized in free-moving mice[36], a three-way connection of catheters were coupled to link a syringe infusion pump for infusion of saline, a pressure transducer for pressure recording, and catheter insertion inside the mouse colon so that intracolonic pressures were recorded during a constant rate of saline infusion into the colon. The polyethylene (PE)−50 intracolonic catheter was placed via anus with the catheter tip 2.5 cm away from the anus. A rate of 1.2 mL/h saline infusion was used throughout for both male and female mice. The pressure transducer was connected to a bridge amplifier and a computer recording system (AD Instrument, Milford, MA) for data acquisition and analysis. Colonometrogram was analyzed accordingly to our published protocols[36]. The colonic mechanosensitivity was presented by changes in the amplitude of intracolonic pressures ($A_{ICP}$) as results of colonic stretch-reflex contraction.

## Cystometry and cystometrogram analysis

The above-mentioned three-way connection setup was also used for cystometrical recording of the urinary bladder function in conscious mice. In a surgical procedure (under 2.5% isoflurane) that exposed the urinary bladder via a midline abdominal incision, the PE-50 intravesical

**Fig. 7 | Piezo2 in nociceptive neurons mediates colonic hypersensitivity and mechanical pain in male mice. a** Piezo2[wt] and Piezo2[cKO] male mice that receive either vehicle (V) or 2,4,6-Trinitrobenzene sulfonic acid (T: TNBS) are subject to colonometrical analysis on day 7 post drug treatment. **b** Average of amplitudes of intracolonic pressures (A$_{ICP}$) from each animal are analyzed. $n = 3$ for vehicle treatment and n = 4 for TNBS treatment of biologically independent mice. Data are presented as mean values +/- SEM. One-way ANOVA with Tukey's multiple comparisons test (Piezo2[wt]-V vs. Piezo2[wt]-T: $p = 0.0123$; Piezo2[wt]-V vs. Piezo2[cKO]-T: $p = 0.0069$; $F = 9.769$). **c** Threshold colorectal distension (CRD) pressures that elicits painful behaviors in colonic inflamed mice. $n = 3$ biologically independent Piezo2[wt] mice and $n = 3$ biologically independent Piezo2[cKO] mice. Data are presented as mean values +/- SEM. Two-tailed unpaired $t$ test ($p = 0.0153$, $t = 4.064$, $F = 1.0$). **d** Calcitonin gene-related peptide (CGRP) immunoreactivity in thoracolumbar

dorsal root ganglia (DRG) of Piezo2[wt] mice that receive sham ($n = 3$ biologically independent mice with 2 DRGs from each totaling 21 sections) or noxious CRD ($n = 3$ biologically independent mice with 2 DRGs each totaling 24 sections). Data are presented as mean values +/- SEM. One-tailed unpaired $t$ test ($p = 0.047$, $t = 2.187$, $F = 3.914$). Scale bar: 50 μm. **e** CGRP immunoreactivity in thoracolumbar DRG of Piezo2[cKO] mice that receive sham ($n = 3$ biologically independent mice with 2 DRGs from each totaling 28 sections) or noxious CRD ($n = 6$ biologically independent mice with 2 DRGs each totaling 48 sections). Data are presented as mean values +/- SEM. One-tailed unpaired $t$ test ($p = 0.2542$, $t = 0.6967$, $F = 5.513$). Scale bar: 50 μm. **f** CGRP immunoreactivity in thoracolumbar spinal cord of Piezo2[wt] and Piezo2[cKO] mice that receive sham or noxious CRD ($n = 2$ mice Piezo2[wt] sham, $n = 3$ mice Piezo2[wt] CRD, $n = 3$ mice Piezo2[cKO] CRD). Scale bar: 150 μm.

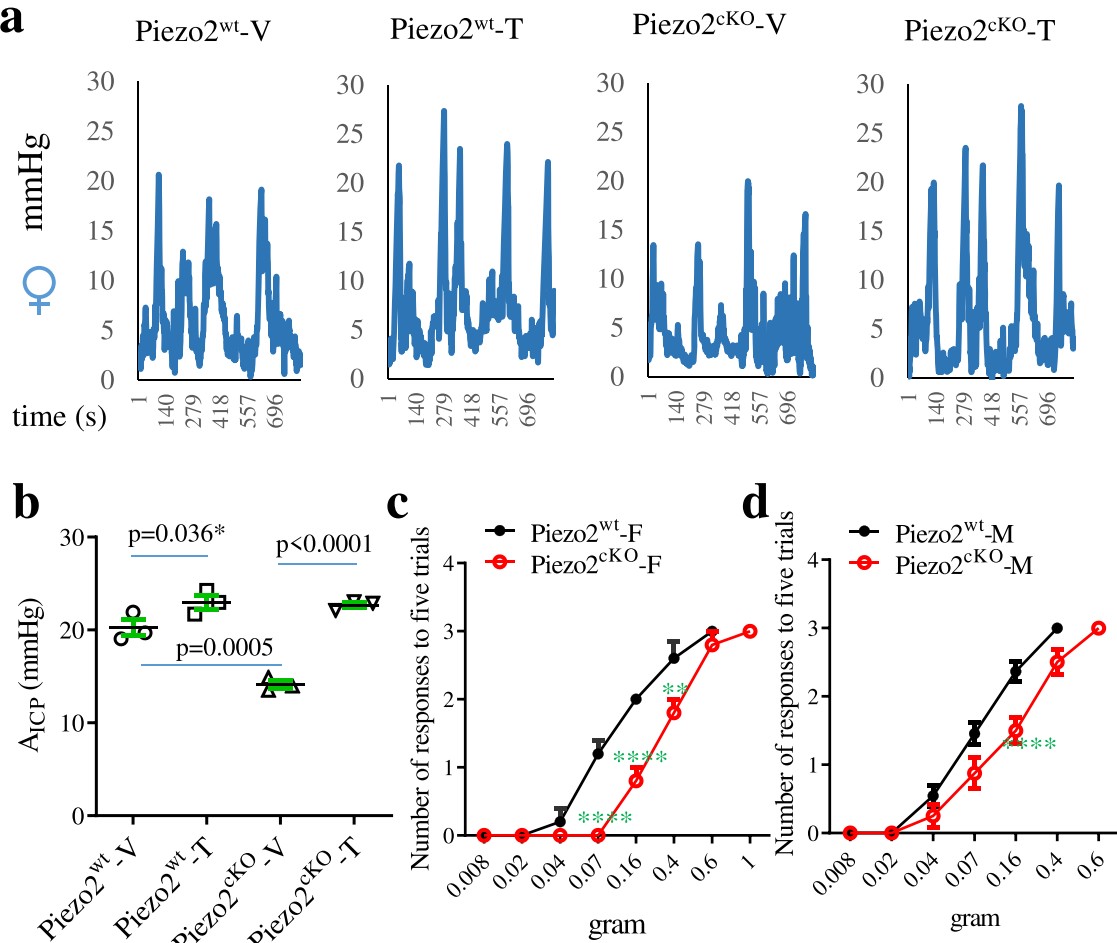

**Fig. 8 | Effects of Piezo2 conditional deletion on female colonic pain hypersensitivity. a** Piezo2[wt] and Piezo2[cKO] female mice that receive either vehicle (V) or 2,4,6-Trinitrobenzene sulfonic acid (T: TNBS) are subject to colonometrical analysis on day 7 post drug treatment. **b** Average of amplitude of intracolonic pressures (A$_{ICP}$) from each animal are analyzed. $n = 3$ biologically independent mice for all groups. Data are presented as mean values +/- SEM. One-way ANOVA with Tukey's multiple comparisons test (Piezo2[wt]-V vs Piezo2[wt]-T: $p = 0.055$, * One-tailed unpaired $t$ test: $p = 0.0364$, $t = 2.42$; Piezo2[wt]-V vs Piezo2[cKO]-V: $p = 0.0005$; Piezo2[cKO]-V vs

Piezo2[cKO]-T: $p < 0.0001$). **c** Hindpaw withdrawal responses to von Frey stimulation of the plantar surface of female Piezo2[wt] ($n = 5$) and Piezo2[cKO] ($n = 5$) biologically independent mice. Data are presented as mean values +/- SEM. Two-way ANOVA with Bonferroni multiple comparisons. ****, $p < 0.0001$; **, $p < 0.01$. **d** Hindpaw withdrawal responses to von Frey stimulation of the plantar surface of male Piezo2[wt] ($n = 11$) and Piezo2[cKO] ($n = 8$) biologically independent mice. Data are presented as mean values +/- SEM. Two-way ANOVA with Bonferroni multiple comparisons. ****, $p < 0.0001$.

catheter with the end flared was inserted into the bladder through a small incision and secured with 6-0 suture at the tip of the bladder dome. The other end of the PE-50 catheter was externalized at the back and connected to the three-way setup for saline infusion and pressure recording. A rate of 1.0 mL/h saline infusion was used. Cystometrogram was analyzed to characterize inter-micturition intervals which was inversely proportional to voiding frequency as indicators of bladder activity.

### DRG neuron single-cell calcium or voltage imaging

DRGs were freshly dissected out and subject to enzymatic disassociation in Gibco Dulbecco's Modified Eagle Medium (DMEM) containing 2 mg/mL collagenase at 32 °C for 1 h. After centrifugation, cells were re-suspended into DMEM containing 10% fetal bovine serum (FBS) and seeded into coated glass-bottom chamber for culture. Cells were fasted for 2-4 h prior to imaging. To apply mechanical stimulation of DRG neurons, we used two different approaches: (1) we poked DRG

neurons using a glass pipette attached to a piezo-driven micro-manipulator with the polished glass probe tip toward the DRG neuron surface at an 80-degree angle. These neurons were pre-incubated with voltage sensor Di-8-ANEPPS (10 μM, Invitrogen™) for 20 min or genetically expressing Archon-1. The voltage changes were detected by alteration in fluorescence intensity that was recorded at a frame rate of 1 kHz (1 frame /per ms)[60]; and (2) we used hypoosmolar solution to stimulate DRG neurons. The hypoosmolar solution (94-194 mosM) has been used to examine the responses of enteric neurons[46]. We tested two sets of hypoosmolar/isotonic solutions on DRG neurons: one set contained 70 mM NaCl (149 mosM) in modified Hanks' Balanced Salt solution (HBSS) or 70 mM NaCl plus 134 mM mannitol in modified HBSS (298 mosM), and another set had 90 mM NaCl without (214 mosM) or with 94 mM mannitol (302 mosM) in modified HBSS. Mannitol was used to bring the hypoosmolar buffer to respective isotonic levels so that the measured responses in neurons were not a result of the alterations of NaCl concentrations. The 70 mM NaCl hypoosmolar solution produced stronger $Ca^{2+}$ responses in DRG neurons therefore was used throughout.

### Cell-type specific mRNA extraction and PCR

Translating ribosome affinity purification (TRAP) technique was used for cell-targeted mRNA extraction from specific DRG neurons[40]. The previously validated DRG mRNA samples from Nav1.8;EGFP::L10a mice[40] were used in this study to characterize gene expression in Nav1.8-expressing nociceptive neurons. The total RNA from the colon and the urinary bladder were extracted using RNAqueous™ Total RNA Isolation Kit (Thermo Fisher Scientific). After reverse transcription, cDNA was subject to conventional PCR and detected by agarose gel electrophoresis. The primers used for PCR were listed in supplementary Table 1.

### Western blot

DRGs were freshly dissected out and subject to homogenization for protein extraction in T-PER buffer (Thermo Fisher Scientific) supplemented with protease inhibitor cocktail (P8340, 1:100, Sigma-Aldrich) and phosphatase inhibitor cocktail 1 (P2850, 1:100, Sigma-Aldrich). After centrifugation at 20,200 g for 10 min at 4 °C, the protein concentration in the supernatant was determined using Bio-Rad DC protein assay kit. Protein extracts were then separated on SDS-PAGE gel and transferred to a nitrocellulose membrane for immunoblotting. The primary antibodies used for western blot were rabbit anti-p-Akt (1:1000, Cell Signaling, Cat# 4060), rabbit anti-Akt (1:1000, Millipore Sigma, Cat# SAB4500800), rabbit anti-p-CREB (1:1000, Cell Signaling, Cat# 9198), and mouse anti-beta actin (1:5000, Sigma, Cat# A2228). The secondary antibodies were anti-rabbit IgG, HRP-linked (1:2500, Cell Signaling, Cat# 7074) and anti-mouse IgG, HRP-linked (1:2500, Cell Signaling, Cat# 7076). The proteins of interests were blotted followed by striping and re-blotting process to visualize the internal controls. The bands were identified by enhanced chemiluminescence (ECL) and analyzed by Imaging J.

### Mechanical stimulation of the distal colon

We used colorectal distention (CRD) to mechanically stimulate the distal colon. To do so, we inserted a mini-balloon coupled to a PE-50 catheter transanally into the distal colon with the center of the balloon located at a position of approximate 2.5 cm away from the anus. The catheter was secured to the mouse tail. The distal end of the catheter was attached to an empty syringe and a sphygmomanometer pressure gauge via a three-way connector to inflate the mini-balloon and record the intra-balloon pressure. This set up was used to (1) induce colonic pain by noxious CRD (80 mmHg inter-balloon pressure), (2) determine threshold pressure at which mice exhibited painful behavioral responses, and (3) measure in vivo calcium transients in whole DRG.

To induce colonic noxious mechanical pain, the procedure was conducted immediately after mice were awake[36]. The intracolonic mini-balloon was rapidly inflated to 80 mmHg and remained for 10 s. The mini-balloon was then quickly deflated to 0 mmHg and remained for 5 s. This inflation/deflation pattern was repeated 3 times. Sham mini-balloon insertion (0 mmHg intra-balloon pressure) served as control.

To determine the CRD threshold pressure that evoked painful behavior, mice were examined when they were fully awake and freely exploring the new environment of the animal enclosure for at least 30 min. During a gradual increment of the CRD pressure, the movements/behaviors of the animal were closely observed. The threshold pressure was recorded at which the animal demonstrated painful behaviors. These behaviors included a sudden immobilization, and/or widening of the hind legs, and/or contraction/tickling of abdominal muscles, followed by attempts to lick/bite the lower abdomen/anus area (manually interfered the biting to protect the catheter). The tests were performed in a blind manner by which one person was assigned to read the behavior of the animals and gave the signal to the person who read and recorded the pressure gauge. Each animal was tested for 3 times with at least 10-min intervals until the animal restored normal behavior to freely explore the chamber. The lowest value (pressure) from the 3 trials for each animal was used.

For in vivo whole DRG $Ca^{2+}$ imaging, we adapted a published protocol that was performed in anesthetized mice[61]. We imaged L2 DRG in response to CRD up to 80 mmHg pressure. The GCaMP mice was placed on a motorized stage under a ZEISS ApoTome fluorescent microscope. After stabilization of the spinal column with a custom designed apparatus, we performed CRD and simultaneous time-lapse recording of $Ca^{2+}$ signals (GCaMP intensity) in DRG during continuous pressure increase. For quantitative analysis, we converted green fluorescence to gray scale and measured the relative intensity using Image J software.

### Somatic mechanical sensitivity

Mice were placed individually into plexiglas chambers sitting above a mesh stand (IITC Life Science Inc., CA) to allow for acclimation to the environment for 30 min. When the animal has all four paws resting on the floor, a von Frey filament was applied perpendicularly to the plantar surface of the hindpaw from underneath of the mesh floor. The mice were each consecutively tested for a filament size, and this was repeated five times before moving on to a filament size of higher force. The tests were blinded to minimize bias. A positive response was considered when a withdrawal behavior (paw licking, shaking, or withdrawal) was noted during or immediately after application. Three positive responses out of the five stimulations were considered as painful responses. Pain threshold was determined by the weight of von Frey filament that evoked three positive responses.

### Gait assay and stride analysis

Regular printing papers were placed under a chamber with a dimension of 50 cm (L) x 5.5 cm (W) x 15.5 cm (H). The hindpaws of mice were gently painted with water-based dyes. Animal was placed to one end of the chamber and freely walked to the other end. The stride length was defined as distance between the adjacent same toe prints. The stride length from the same animal was averaged as one number to be included in the statistical analysis. All procedures and analysis styles stayed the same among all experimental animals.

### Colonic transit assay

Under light anesthesia (1.5% isoflurane), a 3 mm-diameter glass bead (Sigma-Aldrich, St. Louis, MO) was pushed into the distal colon via anus 3 cm inside. A PE-100 catheter with a lightly flared end was used to send the beads. The catheter was then withdrawn and the animal was immediately placed into a glass chamber to recover. The timer started

as soon as the bead was placed in and timer stopped when the bead was expelled out. Animals were tested in the afternoon sessions.

## Drug and virus treatment

Colonic inflammation was induced by intracolonic installation of TNBS (Sigma-Aldrich, St. Louis, MO) as described previously by us[36]. Briefly, a single dose of (TNBS: 75 μL of 12.5 μg/μL TNBS in 30% EtOH) was administered into the mouse colon via a PE-50 catheter through the anus. The proximal tip of the catheter was 2.5 cm inside from the anus. The mouse tail was lifted for 1 min after TNBS installation to avoid drug leakage from the anus. The same amount of 30% EtOH as the vehicle was used in control animals.

CNO treatment was performed either by intraperitoneal route or intrathecal route. For intrathecal injection, an insulin syringe was used to inject drug to a vertebra gap at the lower lumbar position in conscious or lightly anesthetized mice. Tail flick was observed as positive injection.

Adeno-associated viruses were injected intraperitoneally (2 μL stock concentration diluted into 10 μL saline) on postnatal day 5 followed by intrathecal injection (3 μL stock concentration) on week 3. Virus infection of DRGs were examined postmortem for infection efficacy.

## DRG explants culture

DRGs were freshly dissected out and placed into 96-well black wall plate containing DMEM. Each well had one DRG and the segmentally matched DRG pairs were used for comparison between treatment and control. DRGs were settled in cell culture incubator for 2-4 h (fasting) before treatment.

## Flow cytometry

DRGs were digested in digestion media (DMEM, 10 mM HEPES, 5 mg/mL BSA, 100 μg/mL DNase 1) containing 1.6 mg/mL collagenase Type 4 for 30 min at 37 °C. After digestion, cells were filtered through a 40 μm strainer, centrifuged, and washed with FACS buffer (10% fetal bovine serum (FBS), 1 mM EDTA in PBS). Following resuspension in FACS buffer, dead cells were excluded by using live/dead cells labeling Zombie aqua kit (Biolegend # 423101) (Fig. S11). DRGs from Csf1R;GFP mice were directly applied for cell sorting. DRGs from Piezo2[wt] mice were immunostained with Alexa 647-conjugated anti-mouse CD80 (1:200, Biolegend, Cat# 104718, Clone 16-10A1) for 2 h on ice. Nonspecific staining was reduced using CD16/CD32 Fc block antibody (Thermofisher scientific # 14-0161-82). After staining, cells were washed and resuspended with FACS buffer. Cell sorting was performed on a BD Fortessa cell analyzer. Gating FSC-A/SSC-A, FSC-H/FSC-W, SSc-H/SSC-W, and negative for Zombie Aqua defined single, live cell population. FACS data was analyzed using FlowJo software.

## Data and statistical analysis

We used Imaging J, LabChart 8, FlowJo, Zeiss ZEN pro, Nikon NIS-ELEMENTS-BR for data analysis. We used GraphPad Prism 5/9 for statistical analysis. The results from each study were presented as mean ± SEM. Detailed statistical approaches were described in Figure legends. Differences between means at a level of $p \leq 0.05$ were considered to be significant.

## Reporting summary

Further information on research design is available in the Nature Portfolio Reporting Summary linked to this article.

## Data availability

All data were included in this paper. Uncropped original gel scans were attached. Source data are provided in an Excel format.

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

## Acknowledgements

This study was supported by grants NIH R01 DK118137 (LYQ); R01 DK 121131 (LYQ); Virginia's Commonwealth Health Research Board (CHRB) 236-06-18 (LYQ); NIH P30 DA033934 (LYQ). Cell sorting was performed at the Virginia Commonwealth University Flow Cytometry Shared Resource, supported, in part, with funding from NIH-NCI Cancer Center Support Grant P30 CA016059.

## Author contributions

J.M. and C.S. performed and analyzed behavioral experiments. J.M., N.T. and C.S. performed immunostaining. J.M., N.T., C.S., L.Y.Q. analyzed immunostaining. D.S. performed, analyzed, and wrote the part of flow cytometry. L.Y.Q. performed surgery. L.Y.Q., S.S and A.E. performed optogenetic experiments. L.Y.Q., N.T. and S.S. performed calcium imaging. N.T. and L.Y.Q. performed and analyzed mechanical stimulation of DRG neurons. L.Y.Q. and J.M. initiated the project and wrote the manuscript with input from all authors. We thank Dr. Clive Baumgarten for his inputs on applying mechanical stimulation to DRG neurons.

## Competing interests

The authors declare no competing interests.
