## [Peer Review File · Nature Communications]

Piezo2 regulates colonic mechanical sensitivity in a sex specific manner in miceREVIEWER COMMENTS

Reviewer #1 (Remarks to the Author):

Comments to authors on novelty and approach:

The localisation of Piezo channels in the enteric nervous system had previously been studied by Mazzuoli-Weber et al., 2019 (Cell Tissue Res) and they found localisation of both Piezo1 and rarely of Piezo2 in enteric neurons. This paper was a follow-up of Mazzuoli-Weber & Schemann 2015a, where they describe a population of mechanosensitive enteric neurons they called “MEN”. To date, these are the only studies specifically looking at this, please include citations and modify introduction accordingly. Could the authors comment on their findings, compared to those of Mazzuoli-Weber? Could you also comment on the type of colonic afferents you describe as Piezo2+? this would be regarding the neurotransmitter types, etc.

The role of Piezo2 in mechanosensing in the colon has been shown but looking specifically at specialised epithelial cells (enteroendocrine cells). Now this paper investigates colonic afferents from DRG, using retrograde labelling, and successfully showing overlap of Piezo2 in the DRG with those afferents.

However, the imaging of whole mount mucosa is not great quality, compared to mucosal staining done in the past specifically highlighting Piezo2+ cells in EECs. Can the authors please comment on the high level of autofluorescence of the images using the Piezo2-GFP model? A better image of whole mucosa would be ideal and potentially the problematics of working with the Piezo2-GFP model.

The authors focus on the role of DRG Piezo2 and how it influences colonic mechanosensitivity. They do not comment on a link with epithelial Piezo2 or how each can contribute to colonic mechanosensitivity. Previous studies have shown significant differences in colonic motility, secretion, bead expulsion and whole gut transit in epithelial Vil-Piezo2cKOs compared to WT mice. Could authors link those findings to their findings on colonic afferents? (note that there is not a complete inhibition of neither of those physiological responses in epithelial Piezo2cKOs, suggesting enteric neurons would be responsible for the mechanosensitive responses that were not blocked in the epithelial knockouts).

Previous studies had shown differences in colonic motility and secretion in mice lacking mucosal Piezo2, but those studies had not touched on hypersensitivity. The authors use a DREADD approach to specifically activate Piezo2 expressing cells in DRG and find an increase in hypersensitivity in male mice, this being lethal to female mice. The authors explain WT female mice do not respond in the same way to CNO treatment. To further validate these methods, I suggest looking into using a diphtheria-toxin mouse model to evaluate the effects on ablating the population of colonic afferents.

Similarly, using optogenetics, they found that light activation of Piezo2 cells evoked somatic pain in mice. They used neuronal cultures to validate their methods using CNO and light. However, this validation seems incomplete and a bit pointless when they provide only before/after images in Supplementary Figure 4. A correct validation of the model in vitro would be showing the raw traces/data of the calcium imaging showing before/after responses to the stimulation with the LED light, as shown in Supplementary Figure 3 with the CNO experiment. Please provide these.

The authors conclude that chemo genetic and/or optogenetic stimulation of DRG neurons alone expressing Piezo2 is sufficient to evoke mechanical hypersensitivity, which is very interesting and in line with previous studies.

Now, the fact that female mice did not survive CNO treatment specifically due to the expression of Piezo2 in DRG neurons it's a very important finding. The authors also find that the number of Piezo2+ DRG neurons in female is higher and their size as well.

The authors decide to focus on a subpopulation of DRG neurons that also express Nav1.8 (polymodal nociceptors) that are also colonic afferents. Would you please comment on why focusing on this population? Would other populations lead you to different results and did you test another population?

Using this model the aim was to characterise the responses of these cells to "mechanical deformation". Unfortunately, cell swelling is not an acceptable method to assess mechanosensitivity. Cell swelling is extremely damaging for cells, it can activate several intracellular pathways that will rise intracellular Ca²⁺ completely independent of mechanosensors, so it is not reliable to assess the role of Piezo2. Additionally, even though D-GsMTx4 has been shown to be a Piezo inhibitor, other studies have shown it can also block and activate TRP channels, so the effects in osmotic changes could also be non-specific.

The authors explain that previous studies looking into Piezo2 activating currents in nociceptive neurons had been done with a poking device (piezo-transducer driven device and a glass pipette that can poke into the cell with micro to picometer precision in a range from 10-500 ms), which it is currently the gold standard method to activate Piezo2 currents. This method has also been used to study Piezo2 in enterochromaffin cells of the colonic mucosa.

Piezo2 is an ion channel that activates and inactivates within milliseconds, much faster than its sibling receptor Piezo1, with an inactivation constant of around 12 [ms]. It is very important to notice that using the right technique it is crucial in this case, either to record ionic currents or doing calcium imaging. To support their claims, the authors would have to incorporate experiments using the poker approach for calcium imaging or application of shear stress (laminar flow) to unequivocally mechanically-stimulate the cells. I suggest reaching out to potential collaborators if the techniques are not available at your institution.

The authors mention some DRG neurons have been described to contain Piezo1. Their results using a different mouse model show no PCR expression of Piezo1, but ideally, they should show in the same Piezo2-DRG models used earlier what is the expression of Piezo1, with IHC and not only PCR.

Piezo2 deletion seems to have an important effect on female mice, compared to male. Initially with a variation in body weight, slower transit in females and reduced colonic sensitivity. Female mice also had a reduced expression of CGPR. Could the authors explain the relevance of reduced CGPR expression?

Now, female mice seem to have a greater basal colonic mechanical sensitivity, compared to males. Please provide the WT data (not shown?) to support the added comment on this topic on the text, otherwise it seems pointless to mention it.

The authors find that Piezo2 expression is important for the development of colonic inflammation in male mice, as the threshold for CRD to elicit painful behaviour in inflamed mice is smaller in WT mice compared to Piezo2cKO. In Supplementary Figure 8, only qualitative data is provided to support intracellular calcium changes in response to noxious stimuli. Please provide the raw data for this, by analysing the changes in fluorescence and plotting them against time.

Surprisingly, when looking into female mice and colonic hypersensitivity after induction of inflammation, the authors find that inflammation significantly increased mechanical sensitivity in Piezo2cKOs, in contrast with the finding in male knockouts. Similarly, using von Frey hairs, the authors find less differences between WT and KO male mice responses, than those in female, suggesting that conditional deletion of Piezo2 reduces noxious stimuli sensitivity stronger in male mice compared to female.

The provided data strongly suggests a correlation between sex and the role of Piezo2 in mechanical hypersensitivity.

Reviewer #2 (Remarks to the Author):

The manuscript by Madar et al. examines the role of the mechanically-activated ion channel Piezo2 in colonic mechanosensation. Using genetically engineered mouse models they examine how activation of Piezo2-expressing neurons affects colonic mechanical sensitivity. The surprising result reported by the group is that responses in male and female mice are starkly different – in male mice, the interventions evoked colonic mechanical hypersensitivity, but in female caused dyspnea and death. Sexual dimorphism was also observed in the effect of conditional deletion of Piezo2 from nociceptive neurons. The main findings in the manuscript are indeed interesting and novel. However, several important points need to be addressed to ensure robustness and to describe the findings more fully.

Major points

1. An important result in the paper is the astonishing lethality of female mice upon stimulation of Piezo1-expressing cells by either DREADDs. The data should be shown for lethality of female mice – how many mice were tested, how immediate was the lethality, what was the exact cause of death
2. Is there a sexual dimorphism in the expression of Cre in the Piezo2 Cre line? Please also provide the precise genotype of the wt:hM3Dq mice, since they were used as controls in Fig. 1.
3. Why does intrathecal injection of CNO affect breathing of the animals since one would expect that an intrathecal injection should be confined close to the injection site. What is the reason for dyspnea?
4. In figure 1h: what was the sex of the mice used? Was sexual dimorphism seen in the optogenetic stimulation experiments? This is an important point, and data from both male and female mice must be shown.
5. What is the molecular mechanism underlying sexual dimorphism observed?

Minor points

1. The manuscripts needs a thorough read for typos and grammatical errors
2. Additional technical details need to be provided of the injections performed
3. Please mark sex of mice in figures consistently (e.g. Fig 3 includes this but Figs 1,4, 5 do not)
4. Formatting of graphs in Fig 3e and 3f is disrupted

Point-by-Point Responses:

Reviewer #1 (Remarks to the Author):

Comments to authors on novelty and approach:

Comment:

The localisation of Piezo channels in the enteric nervous system had previously been studied by Mazzuoli-Weber et al., 2019 (Cell Tissue Res) and they found localisation of both Piezo1 and rarely of Piezo2 in enteric neurons. This paper was a follow-up of Mazzuoli-Weber & Schemann 2015a, where they describe a population of mechanosensitive enteric neurons they called “MEN”. To date, these are the only studies specifically looking at this, please include citations and modify introduction accordingly. Could the authors comment on their findings, compared to those of Mazzuoli-Weber? Could you also comment on the type of colonic afferents you describe as Piezo2+? this would be regarding the neurotransmitter types, etc.

The role of Piezo2 in mechanosensing in the colon has been shown but looking specifically at specialised epithelial cells (enteroendocrine cells). Now this paper investigates colonic afferents from DRG, using retrograde labelling, and successfully showing overlap of Piezo2 in the DRG with those afferents.

However, the imaging of whole mount mucosa is not great quality, compared to mucosal staining done in the past specifically highlighting Piezo2+ cells in EECs. Can the authors please comment on the high level of autofluorescence of the images using the Piezo2-GFP model? A better image of whole mucosa would be ideal and potentially the problematics of working with the Piezo2-GFP model.

Response to comment: We thank the reviewer for constructive suggestions to strengthen this manuscript. We find that the suggested literatures are very helpful to support our studies. We have cited them and other related papers on MEN in Introduction and discussed them in Results and Discussion. We have compared our findings on Piezo2 in colonic afferent neurons with Dr. Mazzuoli-Weber’s findings on Piezo channels in MEN, along with Piezo2 in mucosa, all of them are important components in colonic mechanosensing with distinct mechanisms. We have performed new experiments to characterize the subtypes of Piezo2+ DRG neurons and found gender-specific differences in the neurotransmitter types between female and male mice (**new Figure 3c-3h**). We agree with the Reviewer on the potential high levels of GFP autofluorescence. We have substituted the figure with mCherry labeling of Piezo2 with GFP as background labeling to show specificity (**Figure 1e**, compare to 1d and 1f showing mCherry specific labeling).

Comment:

The authors focus on the role of DRG Piezo2 and how it influences colonic mechanosensitivity. They do not comment on a link with epithelial Piezo2 or how each can contribute to colonic mechanosensitivity. Previous studies have shown significant differences in colonic motility, secretion, bead expulsion and whole gut transit in epithelial Vil-Piezo2cKOs compared to WT mice. Could authors link those findings to their findings on colonic afferents? (note that there is not a complete inhibition of neither of those physiological responses in epithelial Piezo2cKOs, suggesting enteric neurons would be responsible for the mechanosensitive responses that were not blocked in the epithelial knockouts).

Response to comment: We have included in the Discussion the contribution of Piezo in each of the components, DRG Piezo2, epithelial Piezo2, and enteric Piezo channels, in regulation of colonic mechanosensitivity. Indeed, our study showed that deletion of Piezo2 from nociceptive neurons did not change colonic transit in male mice, suggesting roles of Piezo in the colon (and other non-nociceptive sensory neurons) in regulation of colonic function.

Comment:

Previous studies had shown differences in colonic motility and secretion in mice lacking mucosal Piezo2, but those studies had not touched on hypersensitivity. The authors use a DREADD approach to specifically activate Piezo2 expressing cells in DRG and find an increase in hypersensitivity in male mice, this being lethal to female mice. The authors explain WT female mice do not respond in the same way to CNO treatment. To further validate these methods, I suggest looking into using a diphtheria-toxin mouse model to evaluate the effects on ablating the population of colonic afferents.

Similarly, using optogenetics, they found that light activation of Piezo2 cells evoked somatic pain in mice. They used neuronal cultures to validate their methods using CNO and light. However, this validation seems incomplete and a bit pointless when they provide only before/after images in Supplementary Figure 4. A correct validation of the model in vitro would be showing the raw traces/data of the calcium imaging showing before/after responses to the stimulation with the LED light, as shown in Supplementary Figure 3 with the CNO experiment. Please provide these.

The authors conclude that chemo genetic and/or optogenetic stimulation of DRG neurons alone expressing Piezo2 is sufficient to evoke mechanical hypersensitivity, which is very interesting and in line with previous studies.

Response to comment: We thank the Reviewer for suggestions to ablate the population of colonic afferents using a diphtheria-toxin mouse model to remedy the harm of CNO-induced overall Piezo2 activation to female mice. We have searched two strategies using diphtheria-toxin mouse models: (1) Using Piezo2;DTR (diphtheria toxin A (DTA) receptor) mice and injecting DTA to the colon, however, colonic DTA injection would also ablate Piezo2-expressing cells in epithelia; (2) Injecting retrograde AAV carrying DTA into muscular wall of the colon to retrogradely label colonic afferent neurons. We have tried our best to acquire such virus with no success (It was not available commercially or published by others). Instead, we thought it would be helpful if we could implement an approach that could avoid targeting overall Piezo2-expressing cells (Mice with constitutive ablation of Piezo2 from DRG do not survive beyond 24 hours of birth: Nonomura, K., et al. Nature 541, 176-181, 2017). To do so, we performed new experiments by injecting retrograde AAV-hM3Dq virus to the muscular layer of the colon wall to specifically target Piezo2-expressing colonic afferent neurons (**new Figure S5a, Figure 2c, 2d**). We also performed experiments using localized optogenetic approaches in female mice to target the regions of T13-L2 DRGs where colonic (but not lung) afferent neurons located (**new Figure 2b**). These intersectional modification of Piezo2-expressing colonic afferent neurons did not cause breathing problems in female mice. These results inferred that systemic activation of Piezo2-expressing sensory neurons by CNO that also affected lung primary afferent neurons might underlie the development of dyspnea in female mice. Indeed, studied by Nonomura, K., et al. (Nature 541, 176-181, 2017) showed that optogenetic activation of Piezo2-expressing vagal sensory neurons caused apnoea in adult mice, suggesting that the lung function can be easily disturbed by activation of Piezo2-expressing lung-innervating sensory neurons. We have included the calcium tracing in **Figure S4g**. We thank the Reviewer for the positive feedback on this work to be in line with previous studies.

Comment:

Now, the fact that female mice did not survive CNO treatment specifically due to the expression of Piezo2 in DRG neurons it's a very important finding. The authors also find that the number of Piezo2+ DRG neurons in female is higher and their size as well.

Response to comment: We thank the Reviewer for recognition of the significance of our studies.

Comment:

The authors decide to focus on a subpopulation of DRG neurons that also express Nav1.8 (polymodal nociceptors) that are also colonic afferents. Would you please comment on why focusing on this population? Would other populations lead you to different results and did you test another population?

Using this model the aim was to characterise the responses of these cells to “mechanical deformation”. Unfortunately, cell swelling is not an acceptable method to assess mechanosensitivity. Cell swelling is extremely damaging for cells, it can activate several intracellular pathways that will rise intracellular Ca²⁺ completely independent of mechanosensors, so it is not reliable to assess the role of Piezo2. Additionally, even though D-GsMTx4 has been shown to be a Piezo inhibitor, other studies have shown it can also block and activate TRP channels, so the effects in osmotic changes could also be non-specific.

The authors explain that previous studies looking into Piezo2 activating currents in nociceptive neurons had been done with a poking device (piezo-transducer driven device and a glass pipette that can poke into the cell with micro to picometer precision in a range from 10-500 ms), which it is currently the gold standard method to activate Piezo2 currents. This method has also been used to study Piezo2 in enterochromaffin cells of the colonic mucosa.

Piezo2 is an ion channel that activates and inactivates within milliseconds, much faster than its sibling receptor Piezo1, with an inactivation constant of around 12 [ms]. It is very important to notice that using the right technique it is crucial in this case, either to record ionic currents or doing calcium imaging. To support their claims, the authors would have to incorporate experiments using the poking approach for calcium imaging or application of shear stress (laminar flow) to unequivocally mechanically-stimulate the cells. I suggest reaching out to potential collaborators if the techniques are not available at your institution.

Response to comment: The reasons that we focus on the populations of Nav1.8 nociceptors are that (1) Nav1.8 nociceptors are polymodal and mediate a variety of pain modalities including colonic hypersensitivity, (2) Piezo2 is co-expressed with Nav1.8 in DRG, and (3) our findings demonstrate a sex differential expression and activity of Piezo2 in Nav1.8-expressing DRG neurons (**new Figures 4 and 5**). We have not tested other DRG subpopulations that express Piezo2. Suggested by the Reviewer to incorporate experiments using the poking approach for calcium imaging or application of shear stress (laminar flow) to unequivocally mechanically-stimulate the cells, we established a technique of poking DRG neurons that were loaded with a voltage sensing dye Di-8-ANEPPS. Poking-evoked changes in the intensity of Di-8-ANEPPS fluorescence in DRG neurons were captured by a charged coupled device (CCD) camera with a frame rate of 1kHz (1 frame/ms). We demonstrated that poking elicited action potential within a couple of ms in Piezo2-expressing DRG neurons (**new Figure S6**) and a subpopulation of nociceptive neurons (**new Figure 4e**). We additionally applied this technique to nociceptive neurons that had Piezo2 conditionally deleted (Piezo2^{CKO}, **new Figure 4f**), showing that Piezo2

deletion suppressed poking-elicited activation of a subpopulation of nociceptive neurons (**new Figure 4g**). We agree with the Reviewer that hypoosmolar stimulation-caused cell swelling could damage the cells and could also activate several intracellular pathways that would rise intracellular Ca^{2+} independent of mechanosensors. We also agree that D-GsMTx4 is not a specific inhibitor for Piezo even though it has been shown so. To provide a better control, we re-stimulated the neurons with KCl following hypoosmolar stimulation (**new Figure 5g**) which demonstrated that the neurons were not damaged by our hypoosmolar solution (149 mosM). A similar hypoosmolar stimulation (94-194 mosM) has been used successfully to examine the responses of enteric neurons (Kollmann, P., et al. Submucosal enteric neurons of the cavine distal colon are sensitive to hypoosmolar stimuli. *The Journal of physiology* 598, 5317-5332, 2020). To remedy the non-specificity of D-GsMTx4, we used Piezo2 conditional knockout mice (Piezo2^{CKO}) and genetically matched Piezo2 intact controls (Piezo2^{wt}) in combination with viral-mediated Cre-driven expression of CaMP6s/tdTomato in Piezo2 intact and Piezo2 deleted nociceptive neurons, respectively. Our results showed that Piezo2 mediated, in part, hypoosmolar stimulation-induced activation of nociceptive neurons (**new Figure 5h**). We consulted on Dr. Clive Baumgarten, an expert in mechanical stimulation of a variety of cell types, in preparation and application of isotonic/hypoosmolar solutions and mechanical stimulation strategies to DRG neurons. Of noting, we also tried shear stress in combination with Ca^{2+} imaging on DRG neurons and found that it was inferior to and not as specific as poking or hypoosmolar solution to stimulate DRG neurons.

Comment:

The authors mention some DRG neurons have been described to contain Piezo1. Their results using a different mouse model show no PCR expression of Piezo1, but ideally, they should show in the same Piezo2-DRG models used earlier what is the expression of Piezo1, with IHC and not only PCR.

Response to comment: We would like to clarify that our PCR was nociceptor-targeted PCR that was not on the whole DRG therefore Piezo1 in DRG neurons other than Nav1.8-lineage nociceptors would not be picked up by PCR. As suggested, we performed experiments using a Piezo1 antibody which somehow showed staining on or next to nuclear membrane (**Figure S5d**).

Comment:

Piezo2 deletion seems to have an important effect on female mice, compared to male. Initially with a variation in body weight, slower transit in females and reduced colonic sensitivity. Female mice also had a reduced expression of CGRP. Could the authors explain the relevance of reduced CGRP expression?

Response to comment: CGRP is an excitatory neurotransmitter that is produced in sensory neurons in the periphery. CGRP is largely expressed in Nav1.8-lineage nociceptive neurons. In sensory reflex arc of female mice, it could be possible that Piezo2 conditional deletion from nociceptive neurons (1) reduces terminal mechanical sensing, (2) leads to less activity of nociceptive neurons that further leads to CGRP reduction, and (3) less CGRP release from DRG to the spinal cord could lead to weak sensory-motor reflex of the colon. Our previous studies show that CGRP release to the spinal cord causes an activation of the N-methyl-D-aspartate receptor (NMDAR) in the deep laminae of the thoracolumbar and lumbosacral spinal cord (Kay JC., et al., *Exp Neurol.* 2013 Dec;250:366-75), suggesting an essential role of CGRP in visceral (colon and bladder) sensory reflex pathway.

Comment:

Now, female mice seem to have a greater basal colonic mechanical sensitivity, compared to males. Please provide the WT data (not shown?) to support the added comment on this topic on the text, otherwise it seems pointless to mention it.

Response to comment: We agree with the Reviewer that data from naïve male and female mice do not fit into this study that focuses on Piezo2^{wt} and Piezo2^{cKO} mice. We removed the sentence related to the naïve mice to avoid confusion.

Comment:

The authors find that Piezo2 expression is important for the development of colonic inflammation in male mice, as the threshold for CRD to elicit painful behaviour in inflamed mice is smaller in WT mice compared to Piezo2cKO. In Supplementary Figure 8, only qualitative data is provided to support intracellular calcium changes in response to noxious stimuli. Please provide the raw data for this, by analysing the changes in fluorescence and plotting them against time.

Response to comment: We included Ca²⁺ tracings in the new Figure S9 (**Figure S9a'**, **S9b'**, **S9d'**).

Comment:

Surprisingly, when looking into female mice and colonic hypersensitivity after induction of inflammation, the authors find that inflammation significantly increased mechanical sensitivity in Piezo2cKOs, in contrast with the finding in male knockouts. Similarly, using von Frey hairs, the authors find less differences between WT and KO male mice responses, than those in female, suggesting that conditional deletion of Piezo2 reduces noxious stimuli sensitivity stronger in male mice compared to female.

The provided data strongly suggests a correlation between sex and the role of Piezo2 in mechanical hypersensitivity.

Response to comment: We greatly appreciate the Reviewer for supporting our studies. The constructive suggestions not only give us an opportunity to provide substantial new data to strengthen our findings but also allow us to use this opportunity to develop new approaches and make new discoveries.

Reviewer #2 (Remarks to the Author):**Comment:**

The manuscript by Madar et al. examines the role of the mechanically-activated ion channel Piezo2 in colonic mechanosensation. Using genetically engineered mouse models they examine how activation of Piezo2-expressing neurons affects colonic mechanical sensitivity. The surprising result reported by the group is that responses in male and female mice are starkly different – in male mice, the interventions evoked colonic mechanical hypersensitivity, but in female caused dyspnea and death. Sexual dimorphism was also observed in the effect of conditional deletion of Piezo2 from nociceptive neurons. The main findings in the manuscript are indeed interesting and

novel. However, several important points need to be addressed to ensure robustness and to describe the findings more fully.

Response to comment: We thank the reviewer for the recognition of this work to be novel.

Major points:

Comment:

1. An important result in the paper is the astonishing lethality of female mice upon stimulation of Piezo1-expressing cells by either DREADDs. The data should be shown for lethality of female mice – how many mice were tested, how immediate was the lethality, what was the exact cause of death

Response to comment: For intraperitoneal CNO injection of female Piezo2;hM3Dq mice, we tested 6 mice with descending doses starting from 3 mg/kg body weight as used for male mice. Surprisingly, female mice did not wake up from isoflurane after 3 mg/kg body weight (i.p.) injection. We performed a series of dilution of CNO injection from 1 mg/kg to 10 µg/kg body weight. For intrathecal experiments, we tested a single dose (3 µL of 26 µM CNO solution) that was used for male mice on 3 female mice. Mice that woke up from isoflurane had intense and labored breathing (dyspnea). We observed mice for approximately 2-4 hours after CNO injection. Previous study by Nonomura, K., et al. (Nature 541, 176-181, 2017) showed that optogenetic activation of Piezo2-expressing vagal sensory neurons caused apnoea (where the muscles and soft tissues in the throat relax and collapse sufficiently to cause a total blockage of the airway) in adult mice. Systemic CNO treatment also activates Piezo2-expressing cells regulating the lung, thus it might have similar effects to cause lung dysfunction as optogenetic activation of vagal sensory neurons does to the lung.

Comment:

2. Is there a sexual dimorphism in the expression of Cre in the Piezo2 Cre line? Please also provide the precise genotype of the wt:hM3Dq mice, since they were used as controls in Fig. 1.

Response to comment: We used a reporter to characterize the activity of Cre in Piezo2-Cre line which showed sexual dimorphism (**Figure 2e**). We included wt:hM3Dq mice in the Method section.

Comment:

3. Why does intrathecal injection of CNO affect breathing of the animals since one would expect that an intrathecal injection should be confined close to the injection site. What is the reason for dyspnea?

Response to comment: The upper thoracic DRGs contain primary afferent neurons innervating the lung. Intrathecal CNO injection affects all levels of DRGs that include colonic afferent neurons and lung afferent neurons. As we discussed above, activation of lung afferent neurons can lead to lung dysfunction.

Comment:

4. In figure 1h: what was the sex of the mice used? Was sexual dimorphism seen in the optogenetic stimulation experiments? This is an important point, and data from both male and female mice must be shown.

Response to comment: The original Figure 1h was from male mice. In the revised manuscript, we included results from both male (we moved Figure 1h to new Figure 2a) and female mice (**new Figure 2b**).

Comment:

5. What is the molecular mechanism underlying sexual dimorphism observed?

Response to comment: Our data shows sex differential expression and activity of Piezo2 in whole DRG and nociceptive neurons (**new Figures 3-5**), which could be one of the mechanisms underlying the sexual dimorphic role of Piezo2. However, we do not preclude other underlying mechanisms and Piezo2-mediated pathways.

Minor points

Comment:

1. The manuscripts needs a thorough read for typos and grammatical errors

Response to comment: We have carefully read the manuscript and corrected the grammatical errors.

Comment:

2. Additional technical details need to be provided of the injections performed

Response to comment: We expanded our methodology with more details.

Comment:

3. Please mark sex of mice in figures consistently (e.g. Fig 3 includes this but Figs 1,4, 5 do not)

Response to comment: We presented our data in a consistent format in the revised manuscript.

Comment:

4. Formatting of graphs in Fig 3e and 3f is disrupted

Response to comment: We made corrections to these figures (now Figure 6e and 6f).

We thank the Reviewers for their time and thoughtful comments to strengthen this manuscript. We also thank the Editor for allowing us to provide additional data to support our findings.

REVIEWERS' COMMENTS

Reviewer #1 (Remarks to the Author):

I would like to thank the authors for the big effort they have clearly put into the resubmission of this manuscript. They have added every experiment I suggested in the best of their capacity or performed alternative approaches to support their findings. I am very satisfied with the level of science that has been added to the manuscript. This study is very important and significant, since it highlights a novel role of *piezo2* in colonic hypersensitivity, which is greatly driven by sex. This type of study is crucial for our understanding of sex differences in disease and it will help our field build the foundation for a more personalised and less "male-centric" approach to the treatment of disease.

Reviewer #2 (Remarks to the Author):

The authors have made several changes to the manuscript including new experimental data that greatly strengthens the manuscript. They have adequately addressed my comments.

A couple of minor points emerging from some of the newly included data:

Regarding the di-ANEPPS measurements, the voltage traces recorded are too low to be labeled "action potentials". "Voltage changes" or "Voltage depolarizations" would be better descriptors.

In the abstract, the authors should mention the cause of dyspnea in female mice.

Point-by-Point Responses:

Reviewer #1 (Remarks to the Author):

Comments to authors on novelty and approach:

Comment:

*I would like to thank the authors for the big effort they have clearly put into the resubmission of this manuscript. They have added every experiment I suggested in the best of their capacity or performed alternative approaches to support their findings. I am very satisfied with the level of science that has been added to the manuscript. This study is very important and significant, since it highlights a novel role of *piezo2* in colonic hypersensitivity, which is greatly driven by sex. This type of study is crucial for our understanding of sex differences in disease and it will help our field build the foundation for a more personalised and less "male-centric" approach to the treatment of disease.*

Response to comment: We thank the reviewer for recognition of the significance of our studies and the experiments to support our findings.

Reviewer #2 (Remarks to the Author):

Comment:

The authors have made several changes to the manuscript including new experimental data that greatly strengthens the manuscript. They have adequately addressed my comments.

Response to comment: We thank the reviewer for recognition of our effort in strengthening this manuscript.

Comment:

A couple of minor points emerging from some of the newly included data:

Regarding the di-ANEPPS measurements, the voltage traces recorded are too low to be labeled "action potentials". "Voltage changes" or "Voltage depolarizations" would be better descriptors.

In the abstract, the authors should mention the cause of dyspnea in female mice.

Response to comment: We thank the reviewer for this suggestion. We changed the term "action potentials" to "voltage changes" when presenting the di-ANEPPS measurements. We also included the cause of female dyspnea in the abstract.

In Summary, we thank the Reviewers and Editors for their constructive suggestions toward strengthening this manuscript to meet the publication requirement.